# Attacks on genetic privacy via uploads to genealogical databases

**Michael D Edge[1,2,3]\*, Graham Coop[1,2]\***

[1]Center for Population Biology, University of California, Davis, Davis, United States; [2]Department of Evolution and Ecology, University of California, Davis, Davis, United States; [3]Quantitative and Computational Biology, Department of Biological Sciences, University of Southern California, Los Angeles, United States

**Abstract** Direct-to-consumer (DTC) genetics services are increasingly popular, with tens of millions of customers. Several DTC genealogy services allow users to upload genetic data to search for relatives, identified as people with genomes that share identical by state (IBS) regions. Here, we describe methods by which an adversary can learn database genotypes by uploading multiple datasets. For example, an adversary who uploads approximately 900 genomes could recover at least one allele at SNP sites across up to 82% of the genome of a median person of European ancestries. In databases that detect IBS segments using unphased genotypes, approximately 100 falsified uploads can reveal enough genetic information to allow genome-wide genetic imputation. We provide a proof-of-concept demonstration in the GEDmatch database, and we suggest countermeasures that will prevent the exploits we describe.

**\*For correspondence:**
edgem@usc.edu (MDE);
gmcoop@ucdavis.edu (GC)

**Competing interest:** See
page 19

**Reviewing editor:** Magnus
Nordborg, Austrian Academy of
Sciences, Austria

## Introduction

As genotyping costs have fallen over the last decade, direct-to-consumer (DTC) genetic testing (*Hogarth et al., 2008*; *Hogarth and Saukko, 2017*; *Khan and Mittelman, 2018*) has become a major industry, with over 26 million people enrolled in the databases of the five largest companies (*Regalado, 2019*). One of the major applications of DTC genetics is genetic genealogy. Customers of companies such as 23andMe and Ancestry, once they are genotyped, can view a list of other customers who are likely to be genetic relatives. These putative relatives' full names are often given, and sometimes contact details are given as well. Such genealogical matching services are of interest to people who want to find distant genetic relatives to extend their family tree, and can be particularly useful to people who otherwise may not have information about their genetic relatives, such as adoptees or the biological children of sperm donors. Several genetic genealogy services—including GEDmatch, MyHeritage, FamilyTreeDNA, and LivingDNA (*Table 1*)—allow users to upload their own genetic data if they have been genotyped by another company. These entities generally offer some subset of their services at no charge to uploaders, which helps to grow their databases. Upload services have also been used by law enforcement, with the goal of identifying relatives of the source of a crime-scene sample (*Erlich et al., 2018*; *Edge and Coop, 2019*), prompting discussion about genetic privacy (*Syndercombe Court, 2018*; *Ram et al., 2018*; *Kennett, 2019*; *Scudder et al., 2019*).

The genetic signal used to identify likely genealogical relatives is identity by descent (IBD, [*Browning and Browning, 2012*; *Thompson, 2013*]. We use 'IBD' to indicate both 'identity by descent' and 'identical by descent', depending on context). Pairs of people who share an ancestor in the recent past can share segments of genetic material from that ancestor. The distribution of IBD sharing as a function of genealogical relatedness is well studied (*Donnelly, 1983*; *Huff et al., 2011*; *Browning and Browning, 2012*; *Thompson, 2013*; *Buffalo et al., 2016*; *Conomos et al., 2016*; *Ramstetter et al., 2018*), and DTC genetics entities can use information about the number and length of inferred IBD segments between a pair of people to estimate their likely genealogical

**Table 1.** Key parameters for several genetic genealogy services that allow user uploads as of July 26th, 2019.

| Service | Database size (millions) | Individuals shown | IBS/IBD Segments Reported |
|---|---|---|---|
| GEDmatch | 1.2 | 3000 closest matches shown free; Unlimited w/ $10/month license; any two kits can be searched against each other | Yes if longer than user-set threshold. Min. threshold 0.1 cM, default 7 cM |
| FamilyTreeDNA | 1* | All that share at least one 9 cM block or one 7.69 cM block and 20 total cM | Yes, down to 1 cM, for $19 per kit |
| MyHeritage | 3 | All that share at least one 8 cM block | Yes, down to 6 cM, for $29 per kit or unlimited for $129/year. Customers may opt out |
| LivingDNA | Unknown | Putative relatives out to about 4th-cousin range | Only sum length of matching segments reported |
| DNA.LAND** | 0.159 | Top 50 matches shown with minimum 3 cM segment | Yes |

*Although **Regalado (2019)** reports that FamilyTreeDNA has two million users, he also suggests that only about half of these are genotyped at genome-wide autosomal SNPs, which is in line with other estimates (**Larkin, 2018**).

**DNA.LAND has discontinued genealogical matching for uploaded samples as of July 26th, 2019.

relationship (**Staples et al., 2016**; **Ramstetter et al., 2017**). These shared segments—IBD segments—result in the sharing of a near-identical stretch of chromosome (a shared haplotype). Shared haplotypes can most easily be identified looking for long genomic regions where two people share at least one allele at nearly every locus.

For the rest of the main text, we focus on identical-by-state (IBS) segments, which are genomic runs of (near) identical sequence shared among individuals and can be thought of as a superset of true IBD segments. Very long IBS segments, say over 7 centiMorgans (cM), are likely to be IBD, but shorter IBS segments, say <4 cM, may or may not represent true IBD due to recent sharing—they may instead represent a mosaic of shared ancestry deeper in the past. Many of the algorithms for IBD detection that scale well to large datasets rely principally on detection of long IBS segments, at least as their first step (**Gusev et al., 2009**; **Henn et al., 2012**; **Huang et al., 2014**). We consider the effect on our results of attempting to distinguish IBS and IBD in supplementary material.

Many DTC genetics companies, in addition to sharing a list of putative genealogical relatives, give customers information about their shared IBS with each putative relative, possibly including the number, lengths, and locations of shared genetic segments (*Table 1*). This IBS report may represent substantial information about one's putative relatives—one already has access to one's own genotype, and so knowing the locations of IBS sharing with putative relatives reveals information about those relatives' genotypes in those locations (**He et al., 2014**). Users of genetic genealogy services implicitly or explicitly agree to this kind of genetic information sharing, in which large amounts of genetic information are shared with close biological relatives and small amounts of information are shared with distant relatives.

Here, we consider methods by which it may be possible to compromise the genetic privacy of users of genetic genealogy databases. In particular, we show that for services where genotype data can be directly uploaded by users, many users may be at risk of sharing a substantial proportion of their genome-wide genotypes with any party that is able to upload and combine information about several genotypes. We consider two major tools that might be used by an adversary to reveal genotypes in a genetic genealogy database. One tool available to the adversary is to upload real genotype data or segments of real genotype data. When uploading real genotypes, the information gained comes by virtue of observed sharing between the uploaded genotypes and genotypes in the database (an issue also raised by **Larkin, 2017**). Publicly available genotypes from the 1000Genomes Project (**Abecasis et al., 2012**), Human Genome Diversity Project (**Cann et al., 2002**), OpenSNP project (**Greshake et al., 2014**), or similar initiatives might be uploaded.

A second tool available to the adversary is to upload artificial genetic datasets (**Ney et al., 2018**). In particular, we consider the use of artificial genetic datasets that are tailored to trick algorithms that use a simple, scalable method for IBS detection, that of identifying long segments in which a pair of genomes contains no incompatible homozygous sites (**Henn et al., 2012**; **Huang et al., 2014**). Such artificial datasets can be designed to reveal the genotypes of users at single sites of interest or sufficiently widely spaced sites genome-wide. We describe how a set of a few hundred

artificial datasets could be designed to reveal enough genotype information to allow accurate imputation of common genotypes for every user in the database.

Below, we describe these procedures and illustrate them using either publicly available or artificial data. We show that under some circumstances, many users could be at risk of having their genotypes revealed, either at key positions or at many sites genome-wide. In particular, we show that GEDmatch, as of mid-December 2019, was vulnerable to an attack we term IBS baiting that obtains genotype data via artificial data uploads. Our results are largely complementary to the independent work of *Ney et al. (2020)*, which was first posted publicly within a week of the first public posting of this manuscript on bioRxiv. In the discussion, we consider our work in light of other genetic privacy concerns (*Erlich and Narayanan, 2014*; *Naveed et al., 2015* and the work of *Ney et al., 2020*), and we give some suggested practices that DTC genetics services can adopt to prevent privacy breaches by the techniques described here.

## Results

We describe three general methods for revealing the genotypes of users in genetic genealogy databases that allow uploads. The first, IBS tiling, involves uploading many real genotypes in order to identify genotype information from many regions in many people. The second, IBS probing, involves uploading a dataset containing a long haplotype that includes an allele of interest, creating matches at this locus. Genotypes at other places in the genome are chosen to be unlikely to generate IBS with any user in the database, so matches with the uploaded dataset are likely to be users who carry the allele of interest. The third method, IBS baiting, involves uploading fake datasets with long runs of heterozygosity to induce phase-unaware methods for IBS calling to reveal genotypes.

### IBS tiling

In IBS tiling, the genotype information shared between a target user in the database and each member of a set of comparison genomes is aggregated into potentially substantial information about the target's genotypes. For example, consider a user of European ancestries. She is likely to have some degree of IBS sharing with a large set of people from across Europe (*Ralph and Coop, 2013*) and beyond. If one knows the user's IBS sharing locations with one random person of European ancestries (and the random person's genotype), then one can learn a little about the user's genotype. But if one can upload many people's genotypes for comparison, then one can uncover small proportions of the target user's genotypes from many of the comparison genotypes, eventually uncovering much of the target user's genome by virtue of a 'tiling' of shared IBS with known genotypes (*Figure 1A*). Similar ideas have been suggested with application to IBD-based genotype imputation (*Carmi et al., 2014*).

We consider the amount of IBS tiling possible within a set of publicly available genotypes for 872 people of European origin genotyped at 544,139 sites. We phased the sample using Beagle 5.0 (*Browning and Browning, 2007*) and used Refined IBD software (*Browning and Browning, 2013*) to identify IBS segments (see Materials and methods). In the main text, we include IBS segments that are not particularly likely to be IBD—these are IBS segments returned by Refined IBD with relatively low LOD scores for IBD, between 1 and 3. True IBD segments reveal more than mere IBS segments about shared genotypes because untyped variants (including rare variants) within an IBD segment are likely to be shared. At the same time, mere IBS is sufficient to infer sharing for SNPs that are genotyped within the segment.

*Figure 2* shows the median amount of coverage obtainable by IBS tiling as a function of comparison sample size, imposing various restrictions on the minimum segment length in cM. (For similar results, see Figure 2b of *Carmi et al., 2014* and Figure 2 of *Panoutsopoulou et al., 2014*) Approximately 2.8 Giga base-pairs (Gbp) were covered by IBS segments anywhere in the genome; we take this to be approximately the maximum possible genomic length recoverable by IBS with our SNP set. Using the entire sample (871 genotypes, since the target is left out) and including all called IBS segments >1 cM, the median person has an average of 60% of the maximum length of 2.8 Gbp covered by IBS segments (with the average taken across their two chromosomes), and sites across 82% of this length have at least one of two alleles recoverable by IBS tiling. Increasing the cM threshold required for reporting substantially reduces the amount of IBS tiling. With a cutoff of 3 cM, approximately 6.9% of the median person's genotype information is recoverable, including at least one of two alleles at sites in 11% of the genome. When a more stringent cutoff of 8 cM is used, only 1% of

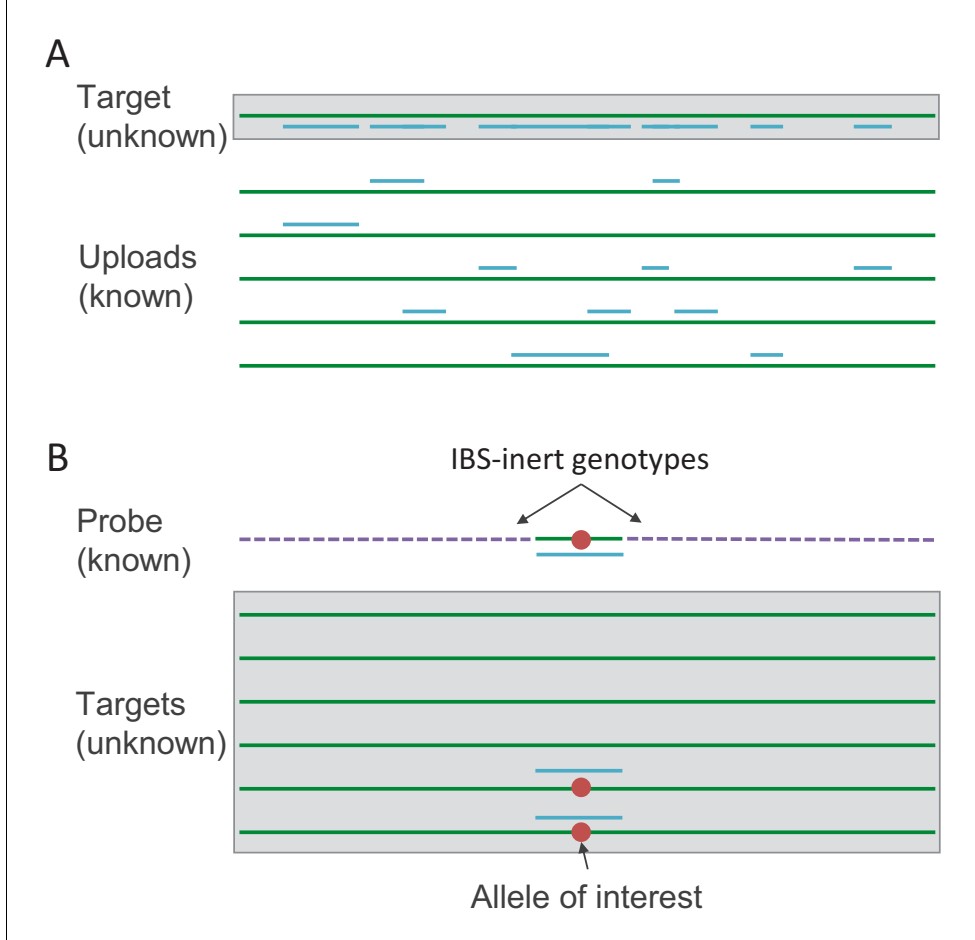

**Figure 1.** Schematics of the IBS tiling and IBS probing procedures. (**A**) In IBS tiling, multiple genotypes are uploaded (green lines) and the positions at which they are IBS with the target (represented by blue lines) are recorded. Once enough datasets have been uploaded, the target will eventually have a considerable proportion of their genome 'tiled' by IBS with uploads that have known genotypes. (**B**) In IBS probing, the uploaded probe consists of a haplotype carrying an allele of interest (red dot) surrounded by 'IBS-inert' segments (purple dashed lines)—fake genotype data designed to be unlikely to share any IBS regions with anyone in the database. In the event of an IBS match in the database, the matching database entry is likely to carry the allele of interest.

the genome has at least one of two alleles recoverable for the median person when using a comparison sample of 871. Our reports for segments longer than 3 cM may be conservative because Refined IBD sometimes splits long IBS segments into multiple shorter segments in the presence of phasing errors (*Browning and Browning, 2013*; *Bjelland et al., 2017*).

For some people, the amount of information obtainable by IBS tiling is even larger. In our sample, the top 10% of people have genotypes across 76% of their total genome covered by IBS tiles, including one or more alleles at sites in at least 93% of the 2.8 Gbp covered by IBS tiles anywhere. If only segments longer than 3 cM are reported, the top 10% of people have one or both alleles covered at sites in at least 38% of the total, and if only segments longer than 8 cM are reported, the top 10% have one or both alleles covered at sites in at least 6% of the total.

The coverage obtained by IBS tiling and its informativeness about target genotypes depends on the specific practices used for reporting IBS information (*Figure 2—figure supplements 1–5*). For example, DTC genealogy services may take additional steps to ensure that any short segments reported are likely to be IBD, not merely IBS. Such steps will tend to decrease the amount of IBS tiling possible, particularly for short segments (*Figure 2—figure supplement 1*). As another example, some DTC genealogy services only report matching segments for pairs of people who share at least one long IBS segment (*Table 1*), but then allow users to see shorter IBS segments (>1cM) for those pairs of

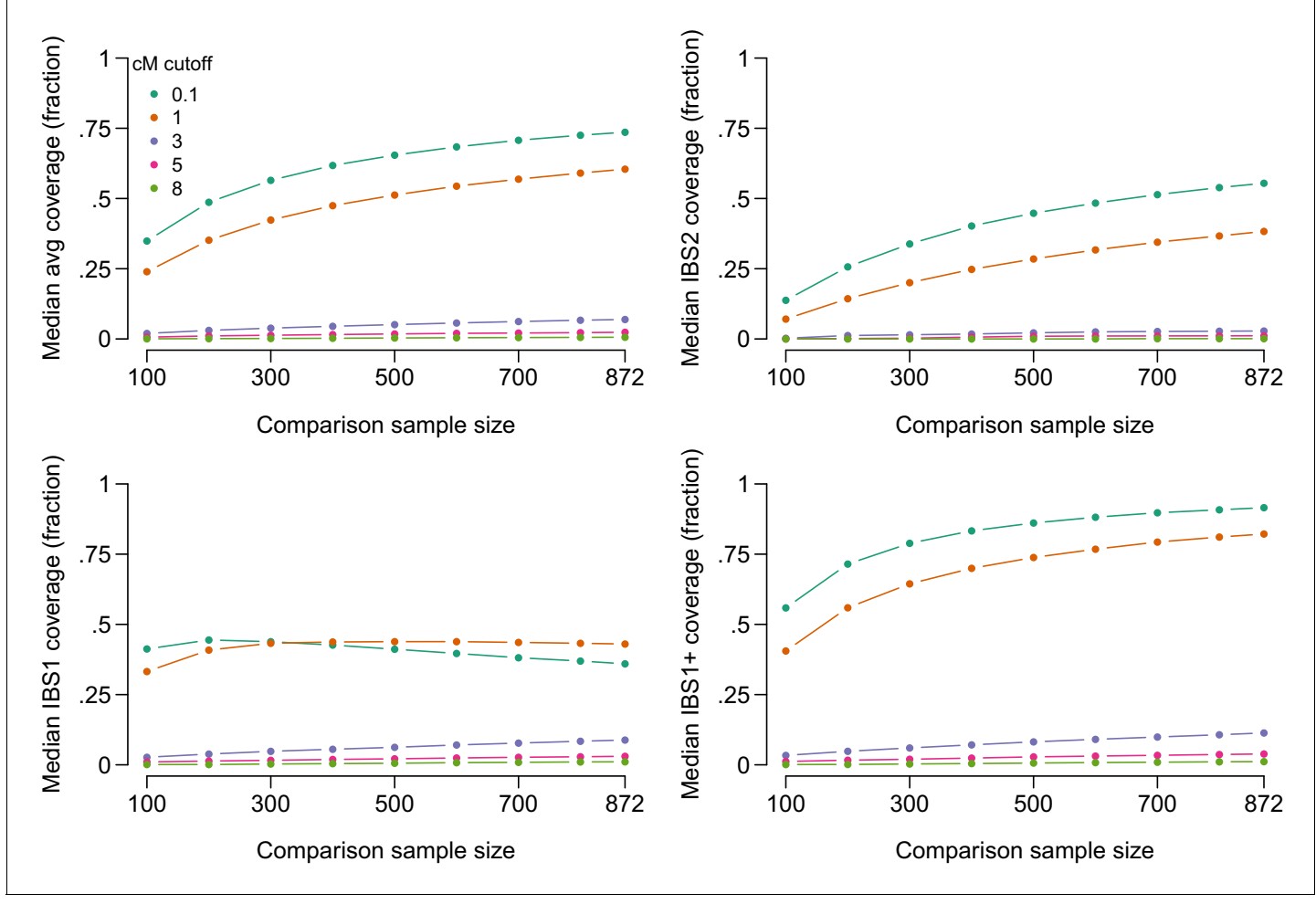

**Figure 2.** Lengths of genome in Giga base-pairs (Gbp) covered by IBS tiling as a function of minimum required length of IBS segments in centiMorgans (cM) and size of a randomly selected comparison sample for the median person in our dataset. The top-left panel shows the average coverage across each of the person's two haplotypes. The top-right shows IBS2 coverage, the length of genome where both haplotypes are covered by IBS tiles. The bottom-left panel shows IBS1, the length of genome where exactly one haplotype is covered by IBS tiles. (IBS1 coverage can decrease at larger comparison sample sizes because IBS2 coverage increases). The bottom-right panel shows IBS1+ coverage, the length of genome covered by either IBS1 or IBS2.

The online version of this article includes the following source data and figure supplement(s) for figure 2:

**Source data 1.** IBS tiling performance.
**Figure supplement 1.** Tiling performance with IBS segments that are unlikely to be IBD filtered out.
**Figure supplement 1—source data 1.** Tiling performance with IBS segments that are unlikely to be IBD filtered out.
**Figure supplement 2.** IBS tiling performance, limiting to comparison samples who share at least 1 IBS segment of 8 cM or more with the target.
**Figure supplement 2—source data 1.** IBS tiling performance, limiting to comparison samples who share at least 1 IBS segment of8cMor more with the target.
**Figure supplement 3.** IBS tiling performance when genotype phasing switches are disallowed.
**Figure supplement 3—source data 1.** IBS tiling performance when genotype phasing switches are disallowed.
**Figure supplement 4.** IBS tiling performance in selected populations.
**Figure supplement 4—source data 1.** IBS tiling performance in selected populations.
**Figure supplement 5.** IBS tiling performance in terms of number of total alleles covered (left panel) and number of minor alleles covered (right panel, 18.6% of total alleles were minor alleles).
**Figure supplement 5—source data 1.** IBS tiling performance in terms of number of total alleles covered (left panel) and number of minor alleles covered.

people. Unsurprisingly, we find that this strategy allows a higher level of IBS tiling than if only long segments are revealed (*Figure 2—figure supplement 2*), because people who share a long IBS segment may also share shorter segments that are hidden if only long segments are reported.

In this demonstration of IBS tiling, we used haplotype information provided by the Refined IBD software to determine which haplotypes were covered by IBS in each person. Most genetic genealogy services that provide information on the location of IBS matches with putative relatives do not provide haplotype information, making it difficult to distinguish IBS1 (in which one chromosome is covered by an IBS segment) and IBS2 (in which both chromosomes are covered by IBS segments). One tool available to an adversary pursuing IBS tiling is to upload genotype information that is homozygous at all sites using one of two phased haplotypes as a basis, effectively searching for IBS with one chromosome at a time. In the presence of phasing errors, some IBS segments may be missed, and the assumption that phase is known would render the coverage rates in *Figure 2* over-estimates. At the same time, the decrease in tiling performance is small for short segments, which can be seen by conducting our test of IBS tiling using Germline software with the haploid flag, which causes putative IBS segments to terminate with a single phasing error (*Figure 2—figure supplement 3*). It may remain difficult to distinguish some cases—such as distinguishing IBS1 from IBS2 with a run of homozygosity on the database genotype—but there will be no question about which uploaded haplotype is IBS with the database genotype. Thus, at any point where a homozygous upload and a target are IBS, at least one of the target's alleles is known. Further, if the target is IBS with any other uploaded datasets at a genetic locus of interest, it will often be possible to infer the target's full genotype.

IBS tiling rates vary somewhat by population, with Finnish samples showing the highest tiling rates among the 1000Genomes populations included (*Figure 2—figure supplement 4*). There also appear to be slight biases for IBS tiles to appear in regions with low SNP density and lower heterozygosity, meaning that the proportion of alleles—and particularly the proportion of minor alleles—recovered by tiling is typically slightly lower than the proportion of the genome length in Mbp covered (*Figure 2—figure supplement 5*).

## IBS probing

IBS probing is an application of the same idea underlying IBS tiling. By IBS probing, one could identify people with specific genotypes of interest, such as risk alleles for Alzheimer's disease (*Corder et al., 1993*), even if the DTC service does not report chromosomal locations of IBS matches. To identify people carrying a particular allele at a locus of interest, one could use haplotypes carrying the allele in publicly available databases. To do so, one would extract a haplotype that surrounds the allele of interest and place it into a false genetic dataset designed to have no long IBS segments with any real genomes (*Figure 1B*). Thus, any returned putative relatives must match at the allele of interest, revealing that they carry the allele. We call this attack 'IBS probing' by analogy with hybridization probes, as the genuine haplotype around the allele of interest acts as a probe. Whereas IBS tiling recovers genetic information from across the genome, IBS probing acts only on a single locus of interest. The advantage is that IBS probing is possible even in databases that do not report the chromosomal locations of IBS segments.

There are several ways of generating chromosomes unlikely to have long shared segments with any entries in the database. One simple way is to sample alleles at each locus in proportion to their frequencies. Chromosomes generated in this way are free of linkage disequilibrium (LD) and thus unlike genuine chromosomes. If the database distinguishes between IBS and IBD, then these fake data are unlikely to register as IBD with any genuine haplotypes. However, they may appear as IBS in segments where genetic diversity is low, depending on the length threshold used by the database. Near-zero rates of IBS can be obtained by generating more unusual-looking fake data, such as by sampling alleles from one minus their frequency or by generating a dataset of all minor alleles.

*Figure 3* shows a demonstration of IBS probing performance in our set of 872 Europeans in a window around the APOE locus. For a 1-cM threshold for reporting IBS, we generated probes by retaining 1.9 cM of real data around a site of interest in the APOE locus from all 872 people. Outside that 1.9-cM window, we generated data by drawing alleles randomly (see Materials and methods). For a 3-cM threshold for reporting IBS, we generated probes by retaining 5.9 cM of real data around the site of interest. With 1-cM matching, 1497 of 1744 haplotypes (86%) matched one of the probes at the site of interest. (Target haplotypes were not allowed to match probes constructed from the

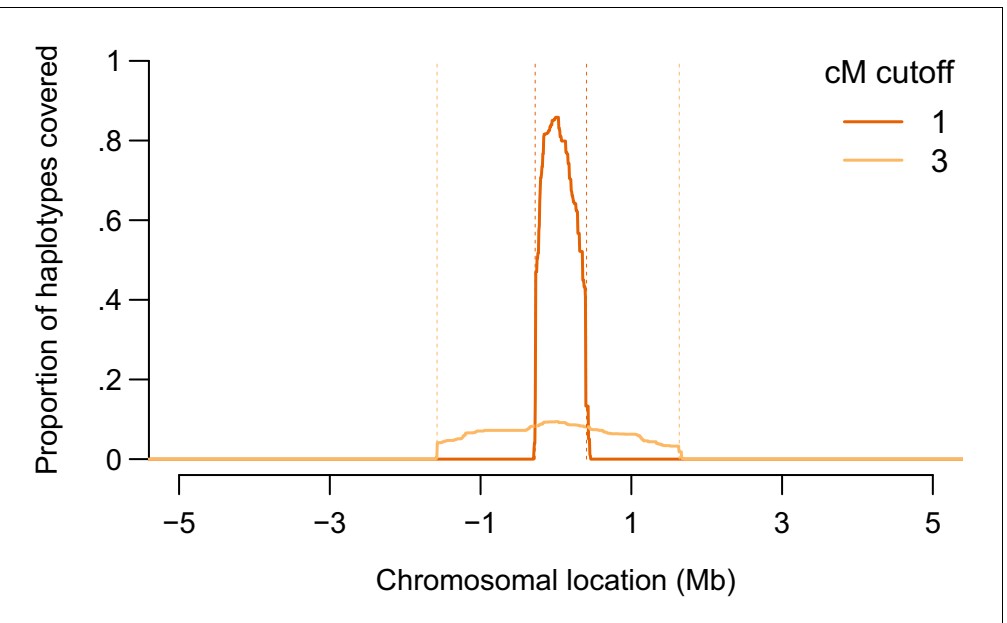

**Figure 3.** A demonstration of the IBS probing method around position 45411941 on chromosome 19 (GRCh37 coordinates), in the APOE locus. We show the proportion of haplotypes among the 872 Europeans in our sample covered IBS by probes constructed from the sample, as a function of the chromosomal location in a 10-Mb region around the site of interest. In red, we show the coverage using a 1-cM threshold for reporting IBS, where the probes are constructed using real data in a 1.9-cM region centered on the site of interest (region boundaries shown in dashed orange). In orange, we show the coverage using a 3-cM threshold for reporting IBS, where the probes are constructed using real data in a 5.9-cM region around the site of interest.
The online version of this article includes the following figure supplement(s) for figure 3:

**Figure supplement 1.** IBS probing with including only segments with LOD>3.
**Figure supplement 2.** IBS probing using Germline (*Gusev et al., 2009*) in haploid mode.

same person that carried the target haplotype). With 3-cM matching, 164 of 1744 haplotypes (9.4%) matched one of the probes at the site of interest. Very few matches occurred outside the region of interest—none with a 3-cM threshold and only 0.1% of matches with a 1-cM threshold. Moreover, we generated different inert genotypes for all 872 probes, and the great majority of these had no matches with any real sample. An adversary would only need to generate one inert dataset, which can be tested by uploading to the database and confirming that no matches are returned. Probes could then be constructed by stitching real haplotypes at the site of interest into the the same set of inert data. The probes would then be likely to match each other, but the adversary would know those identities and could ignore those matches.

The efficacy of IBS probing will depend on the minimum IBS-match length reported to users, the specific methods used for identifying IBS segments (*Figure 3—figure supplements 1–2*), and whether the genotype of interest is included on the SNP chip. These factors vary in terms of whether they affect the sensitivity of IBS probing—the proportion of people carrying the allele of interest returned by a probe or set of probes—or the precision of IBS probing—the proportion of people returned by a probe who in fact carry the genotype of interest. For example, high thresholds for IBS reporting will mean that uploaded genotypes will need to have long IBS segments with targets at the locus of interest. Long IBS segments are likely to represent relatively close genealogical relatives (i.e. long IBS segments are likely to be IBD segments), and not many targets will be close relatives of the source of any given haplotype of interest, meaning that the sensitivity of IBS probing is reduced by reporting thresholds that require long IBS segments. If the locus of interest or a highly correlated one is not included on the chip used to genotype either the uploaded sample or the target sample, then probing may only expected to work well if the upload and the target are truly IBD rather than merely IBS, reducing the precision of IBS probing for variants that are not genotyped. Limiting

probing results to likely IBD matches will decrease the number of matches returned, particularly for short cM thresholds (*Figure 3—figure supplement 1*).

Another factor that will affect the success of IBS probing is the frequency of the allele of interest. For example, if the allele of interest is very rare, then it is likely to be only somewhat enriched on the haplotypes that tend to carry it, and reported matches may not actually carry the allele, even if they are IBD with an uploaded haplotype that carries it. IBS probing will perhaps be most sensitive and precise when the allele of interest is both common and relatively young, as is the case for founder mutations. In this case, most carriers of the allele will share the same long haplotype around the site of interest, meaning that fewer probes would need to be uploaded in order to learn the identities of the majority of the carriers in the database.

## IBS baiting

IBS tiling and IBS probing take advantage of publicly available genotype data. The idea of both is that an adversary uploads genuine genetic datasets—or, in the case of IBS probing, datasets with genuine segments—to learn about entries in the database that share segments with the uploaded genomes.

In this section, we describe an exploit called IBS baiting. The specific strategy for IBS baiting that we describe is possible if the database identifies putative IBS segments by searching for long regions where a pair of people has no incompatible homozygous sites. An incompatible homozygous site is a site at which one person in the pair is homozygous for one allele, and the other person is homozygous for the other allele. Identifying IBS segments in this way does not require phased genotypes and scales relatively easily to large datasets—we refer to methods in this class as 'phase-unaware' and contrast them with phase-aware methods for IBS detection. Phase-unaware methods are robust to phasing errors, which are an issue for long IBD segments (*Durand et al., 2014*). Major DTC genetics companies have used phase-unaware methods in the past for IBS detection (*Henn et al., 2012*; *Hon et al., 2013*), and some state-of-the-art IBD detection and phasing pipelines feature an initial phase-unaware step (*Huang et al., 2014*; *Loh et al., 2016*).

The main tool used in IBS baiting is the construction of apparently IBS segments by assigning every uploaded site in the region to be heterozygous. (SNPs with missing data may also be included in these regions). These runs of heterozygosity, which are unlikely to occur naturally (unlike runs of homozygosity, [*McQuillan et al., 2008*; *Pemberton et al., 2012*]), will be identified as IBS with every genome in the database using phase-unaware methods: because they contain no homozygous sites at all, they cannot contain homozygous sites incompatible with any person in the database.

Here, we consider a database in which an apparent IBS segment is halted exactly at the places at which the first incompatible homozygous site occurs on each side of the segment. We also assume that the database detects all segments without incompatible homozygous sites that pass the required length threshold. *Ney et al. (2020)* independently proposed a similar approach in their section VII 'Genetic Marker Extraction Using Matching Segments,' showing that GEDmatch was vulnerable to it. Similarly, we demonstrate below that IBS baiting can be implemented against GEDmatch.

### Single-site IBS baiting

The simplest application of IBS baiting is to use it to reveal genotypes at a single site. If IBS is identified by looking for single incompatible homozygous sites and missing data can be ignored, then users' genotypes at any single biallelic site of interest can be determined by examining their putative IBS with each of two artificial datasets (*Figure 4A*). In each artificial dataset, the site of interest is flanked by a run of heterozygosity. The combined length of these two runs of heterozygosity must exceed the minimum length of IBS segment reported by the database. The adversary uploads two datasets with these runs of heterozygosity in place. In one dataset, the site of interest is homozygous for the major allele, and in the other, the site of interest is homozygous for the minor allele. If the target user is homozygous at the site of interest, then one of these two uploads will not show a single, uninterrupted IBS segment—IBS will be interrupted at the site of interest (or may not be called at all). If the IBS segment with the dataset homozygous for the major allele is interrupted, then the target user is homozygous for the minor allele. Similarly, if the IBS segment with the dataset homozygous for the minor allele is interrupted, then the target user is homozygous for the major allele. If both uploads show uninterrupted IBS segments with the target, then the target user is heterozygous at the site of interest. Thus, for any genotyped biallelic site of interest, the genotypes of every user

shown as a match can be revealed after uploading two artificial datasets. Depending on how possible matches are made accessible to the adversary, the genotypes of every user could be returned. Genotypes of medical interest that are often included in SNP chips, such as those in the APOE locus (*Corder et al., 1993*), are potentially vulnerable to single-site IBS baiting.

Here, we have considered a database using the simplest possible version of a phase-unaware method for detecting IBS, that in which an apparent IBS segment is halted exactly at the places at which the first incompatible homozygous site occurs on each side of the segment. In principle, phase-unaware IBS-detection algorithms can be altered to allow for occasional incompatible homozygous sites before halting as an allowance for genotyping error, or the extent of the reported region might be modified to be less than the full range between incompatible homozygous sites. Versions of IBS baiting might be developed to work within such modifications. The key insight is that if two artificial kits differ at exactly one site in a region and they produce two different patterns of called IBS with a target, then the target's genotype is revealed at that site. For example, if a database uses a phase-unaware method for IBS calling that requires two incompatible homozygous sites before a putative IBS segment is halted, then an attacker might modify our scheme by putting in a rare homozygote at a site near the key site. For most target users, the rare homozygote in the uploaded files would be an incompatible homozygous site, implying that a mismatch at the key site will cause a break in a putative IBS region. By using different homozygote genotypes nearby, an attacker might still identify the genotypes of everyone in the database at the key site. As discussed below, such measures do not appear to be necessary to perform IBS baiting in GEDmatch. Further, in GEDmatch, uploading a third bait dataset with a missing genotype at the key site can distinguish targets with missing genotypes from heterozygous targets.

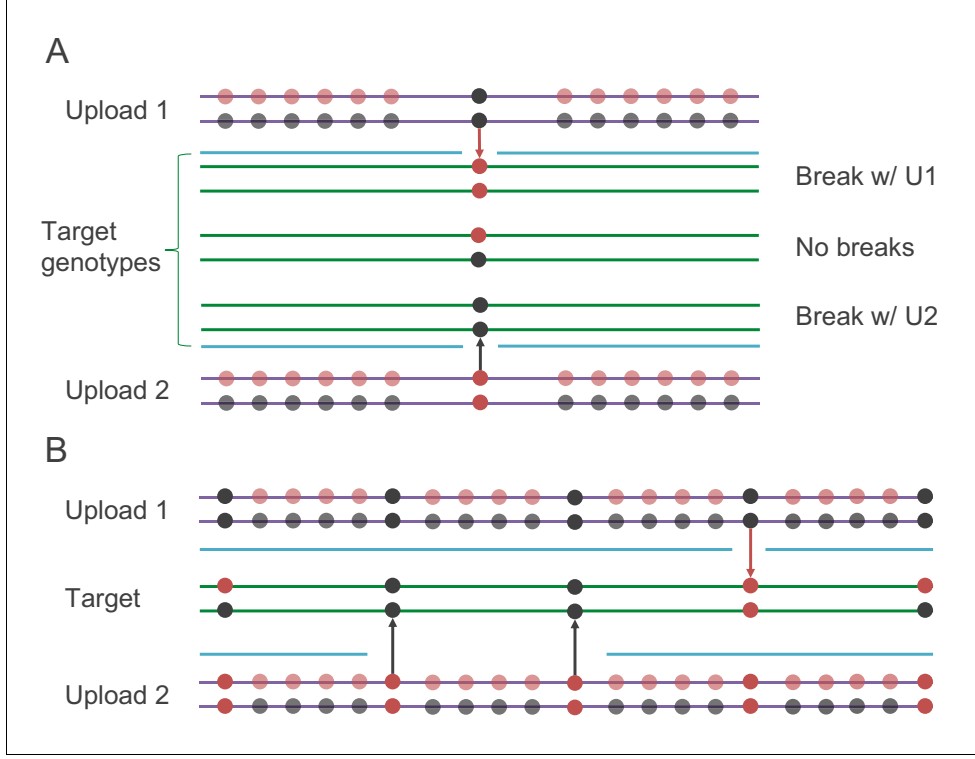

**Figure 4.** Schematics of the IBS baiting procedure. (**A**) To perform IBS baiting at a single site, two uploads are required, each with runs of heterozygous genotypes flanking the key site. At the key site, the two uploaded datasets are homozygous for different alleles. The three possible target genotypes at the key site can each be determined by examining their IBS coverage with the uploads. If there is a break in IBS with either upload, then the target is homozygous for the allele not carried by the upload that shows the break in IBS (with the broken IBS segment shown as a cyan line). If there is no break in IBS with either upload, then the target is heterozygous at the key site. (**B**) Target genotypes at many key sites across the genome can be learned by comparison with two uploaded datasets, as long as key sites are spaced widely enough.

Single-site IBS baiting could also be used if chromosomal locations of matches are not reported. To do so, one would use the the scheme we describe in a large region surrounding the locus of interest and use fake IBS-inert segments to fill in the rest of the dataset.

## Parallel IBS baiting

The second method we consider applies the IBS baiting technique to many sites in parallel (*Figure 4B*). By parallel application of IBS baiting, users' genotypes at hundreds or thousands of sites across the genome can be identified by comparison with each pair of artificial genotypes. By repeated parallel IBS baiting, eventually enough genotypes can be learned that genotype imputation becomes accurate, and genome-wide genotypes could in principle be imputed for every user in the database. If IBS segments as short as 1 cM are reported to the user, then accurate imputation (97–98% accuracy) becomes possible after comparison with only about 100 uploaded datasets. The procedure starts by designing a single pair of uploaded files as follows:

1. Identify a set of key sites to be revealed by the IBS baiting procedure. For every key site, the sum of the distances in cM to the nearest neighboring key site on each side (or the end of the chromosome, if there is no flanking key site on one side) must be at least the minimum IBS length reported by the database.
2. Produce two artificial genetic datasets. In each, every non-key site is heterozygous. In one, each key site is homozygous for the major allele, in the other, each key site is homozygous for the minor allele.
3. Upload each artificial dataset and compare them to a target user. Key sites that are covered by putative IBS segments between the target and both artificial datasets are heterozygous in the target. The target is homozygous for the major allele at key sites that are covered by putative IBS segments between the target and the major-allele-homozygous dataset only. Similarly, the target is homozygous for the minor allele at key sites that are covered by putative IBS segments between the target and the minor-allele-homozygous dataset only.

Carrying out this procedure reveals the target's genotype at every key site. If IBS segments of length at least $t$ cM are reported, and a chromosome is $c$ cM long, then up to $2c/t - 1$ key sites can be revealed with each pair of uploaded files. (To see this, consider the case where $c = tk$, with $k$ a positive integer, and place key sites at $t/2, t, 3t/2, ..., c - t/2$. This calculation ignores the possibility of missing data at key sites in the target). This means that with a minimum reported IBS threshold of 1 cM, 100 uploaded datasets could reveal approximately 100 genotypes per cM, which is enough to impute genome-wide genotypes at 97 - 98% accuracy (*Shi et al., 2018*). In principle, the key sites could also be chosen to ensure good LD coverage and higher imputation accuracy. Of course, higher accuracy imputation can be obtained by recovering exact genotypes for more sites, and with several thousand uploads, the genotypes at every genotyped site could be revealed by IBS baiting without the need to impute.

## IBS baiting in GEDmatch

We hypothesized that IBS baiting would work in the GEDmatch DTC database. GEDmatch provides no public documentation of the IBS algorithm they use, but IBS segments identified by GEDMatch seem to terminate only on incompatible homozygous sites, as would be expected if they use phase-unaware IBS detection. Specifically, the GEDmatch 1-to-1 match tool identifies the locations of IBS segments between pairs of genetic datasets ('kits' in GEDMatch terminology) and allows the user to specify the minimum genetic length and minimum number of matching SNPs to include in a segment. The 1-to-1 tool also returns a 'full resolution' picture of the chromosome that appears to be a SNP-by-SNP picture of the match between the kits along each chromosome. (These pictures are themselves a major security risk. We alerted GEDmatch to the risk in a July 24th email (posted here: https://github.com/mdedge/IBS_privacy/blob/master/IBS_baiting_demo/GEDmatch_emails.pdf) but did not analyze them further. *Ney et al. (2020)* showed in detail that the images provided by GEDmatch allow an adversary to learn the full genotype of a target person).

To demonstrate IBS baiting in GEDmatch, we uploaded a small number of artificial genotypes to their database beginning in late November 2019. These kits were designed in accordance with the algorithm discussed above, but with some slight alterations to bypass counter-measures that GEDMatch has put in place since we (and, independently, Ney and colleagues) informed them of the risk

of IBS baiting in summer 2019. Before uploading any data to GEDmatch, we first confirmed our planned procedure with the UC Davis IRB and with GEDmatch representatives. We uploaded our kits into the GEDmatch 'research' and not 'public' category to prevent matches to the public database, and only used the 1-to-1 IBS match tool among our own uploaded test kits. In this way, we avoided interacting with any genotype data of real GEDmatch users and did not violate GEDmatch's terms and conditions.

We targeted four random SNPs along chromosome 22 for IBS baiting. We uploaded two bait genotype kits (B1 and B2) that had opposite-homozygote genotypes at each of these key SNPs. Each key SNP was in turn surrounded by a ~1cM stretch of SNPs containing genotypes that were either heterozygoous or coded missing. The rest of the genome was specified to be IBS-inert. We then uploaded three target genotype datasets whose genotypes we wanted to determine at the key sites. Two of these target kits (T1 and T3) had opposite-homozygous genotypes at each of the key SNPs, while the third (T2) was heterozygous at each key SNP. (See subsection 'GEDmatch demonstration' in the Materials and methods for more details on the kit design). We then used the GEDmatch 1-to-1 match tool, choosing the parameters so a single opposite-homozygous genotype between a bait and target kit would interrupt a putative IBS segment.

In each case, our two bait kits had the correct IBS patterns with the target kits, allowing correct determination of the target genotypes by IBS baiting. On the left of *Figure 5*, we show a zoomed-in view of the three targets' matches around one of the key SNP sites. The homozygous targets have IBS matches with only one of the bait kits, whereas the heterozygous target has IBS matches with both bait kits. This pattern is seen across all four target regions (right side of *Figure 5*, see section 'GEDmatch demonstration' of the Materials and methods for more detailed results). The target and bait kits displayed in *Figure 5* were uploaded and analyzed on December 15, 2019, showing that GEDmatch has remained vulnerable to IBS-baiting attacks even after its acquisition by Verogen, which was announced on December 9, 2019.

## Discussion

We have suggested several methods by which an adversary might learn the genotypes of people included in a genetic genealogy database that allows uploads. Our methods take advantage of both the population-genetic distributions of IBS segments and of methods used for calling IBS. In particular, IBS tiling works simply because there are background levels of IBS (and IBD) even among distantly related members of a population (e.g. *Ralph and Coop, 2013*). In our dataset, the median person had the majority of their genetic information susceptible to IBS tiling on the basis of other members of the dataset, depending on the procedures used for reporting IBS. IBS tiling performance will also depend on the ancestries of the target and comparison samples because IBD rates differ within and among populations (*Palamara et al., 2012*; *Carmi et al., 2013*; *Ralph and Coop, 2013*), as well as on the prevalence of close biological relatives in the dataset. IBS tiling performance improves as the size of the comparison sample increases. Thus, if enough genomes are compared with a target for IBS, eventually a substantial amount of the target genome is covered by IBS with one or more of the comparison genomes.

IBS probing combines the principles behind IBS tiling with the idea of 'IBS-inert' artificial segments. If the majority of the genome—everywhere except a locus of interest—can be replaced with artificial segments that will not have IBS with any genome in the database, then the adversary knows that any matches identified are in a locus of interest. As such, IBS probing could be used to reveal sensitive genetic information about database participants even if chromosomal locations of matches are not reported to users.

Finally, IBS baiting exploits phase-unaware IBS calling algorithms that use incompatible homozygous sites to delimit putative IBS regions. Although such methods can be useful in genetic genealogy because they scale well to large data, they are vulnerable to fake datasets that include runs of heterozygous sites, which will be identified as IBS with everyone in the database. By inserting homozygous genotypes at key sites and heterozygotes everywhere else, we estimate that approximately 100 well-designed uploads could reveal enough genotypes to impute genome-wide information for any user in a database, provided that the threshold for reporting a matching segment is approximately 1 cM. Similarly, two uploads could reveal any genotype at a single site of interest, such as

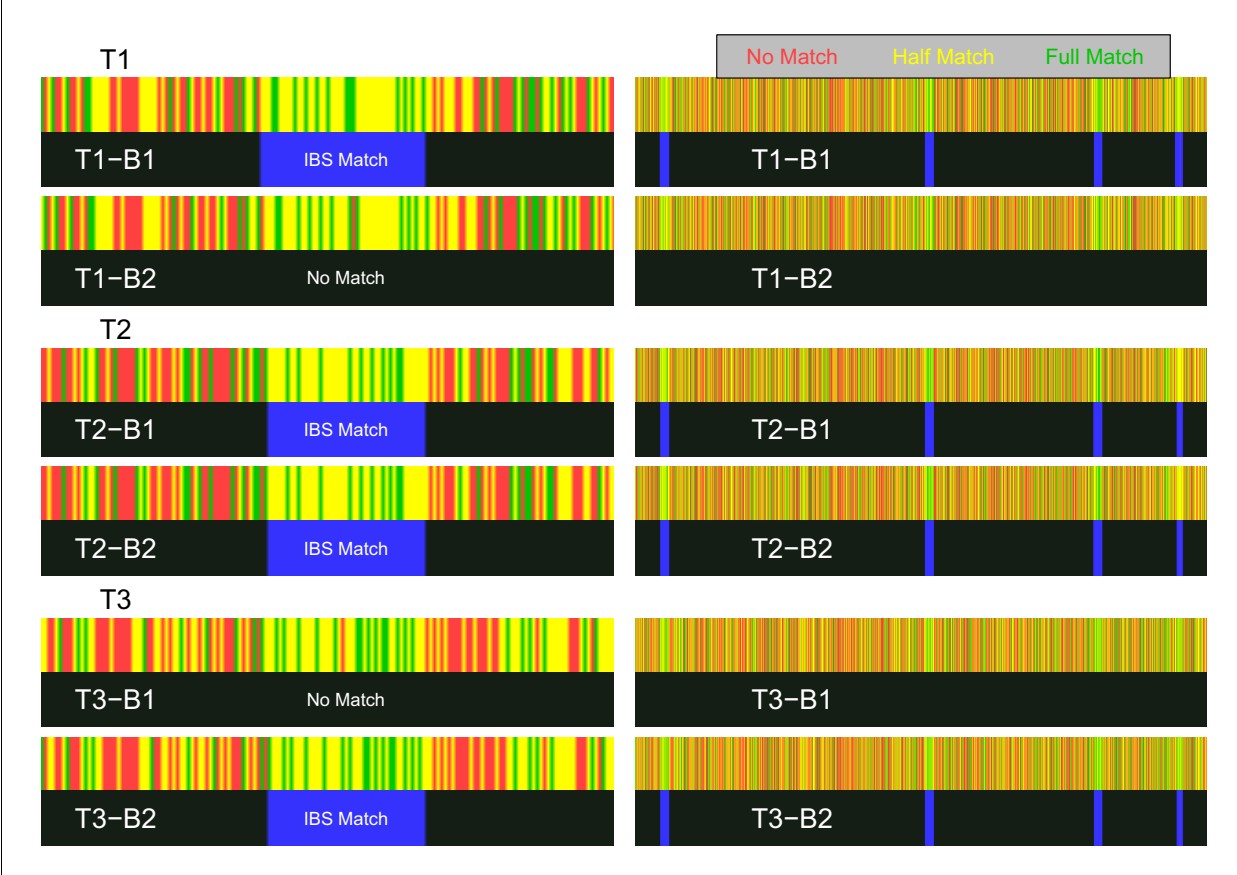

**Figure 5.** Visualization of IBS baiting using GEDmatch's 1-to-1 chromosome browser. Left: Zoomed-in view of the region containing key SNP 1, showing the three target kits (T1–T3) matched to the two bait kits (**B1 and B2**). Right: Zoomed-out views of regions containing all four key SNPs on chromosome 22. For each pair of bait and target kits, the top rectangle (red, yellow, or green) shows the GEDmatch SNP-level pairwise genotype-match image (colored to show no match, half match, or full match) returned by the 1-to-1 GEDmatch tool. The bottom rectangle (black and blue) shows the GEDmatch IBD-track image, black for no putative IBD match, blue regions showing putative IBD segments. The white text on the IBS track is not provided by GEDmatch and was added as a guide to the eye. Opposite-homozygote calls at the key SNP are seen in the left panel as a red line in an otherwise matching region (yellow and green). The spatial positions of SNPs in the match panel appears to have been jittered; for example the location of the red line varies slightly in the different plots that should have the same coordinate system (perhaps as a countermeasure against a *Ney et al., 2020*-style attack).

rs429358, which reveals whether the user carries an APOE-ε4 variant and is associated with risk of late-onset Alzheimer's disease.

There are millions of people enrolled in genetic genealogy databases that allow uploads (*Table 1*). Genetic genealogy has many applications, and uploads are popular with users who want to find relatives who may be scattered across different databases. Though allowing uploads brings several benefits for both customers and DTC services, it also entails additional privacy risks. Users of DTC genetic genealogy services that allow uploads could be at risk of having their genetic information extracted by the procedures we describe here, depending on the methods that these services use to identify and report IBS. Concerns arising from the methods we report are in addition to standard digital security concerns. The attacks we describe require little special expertise in computing; the adversary only needs to be able to procure or create the appropriate data files and to process and aggregate the information returned from the database.

We have not set out to determine precisely how vulnerable users of each specific DTC service are. We do not know the full details of methods used by each service for matching, nor have we attempted to deanonymize any real users' genotypes. We contacted representatives of each of the organizations listed in *Table 1* 90 days (July 24th, 2019) before posting this manuscript publicly in order to give them time to repair any security vulnerabilities related to the methods we describe.

We have posted our emails to GEDmatch representatives here: https://github.com/mdedge/IBS_privacy/blob/master/IBS_baiting_demo/GEDmatch_emails.pdf.

On the basis of our results, we do have serious concerns about the privacy of GEDmatch users. As of this writing, GEDmatch uses length thresholds for displaying matching segments that are too short, allowing for effective IBS tiling attacks, and GEDmatch also appears to use phase-unaware IBD detection methods, allowing for IBS baiting attacks. Additionally, as detailed by *Ney et al. (2020)*, whose work was independent of ours, GEDmatch provides users with high-resolution images comparing the chromosomes of any two users at SNP-level resolution, allowing for reconstruction of a target's genotype using these images. GEDmatch was recently purchased by Verogen, a forensic genetics company, but as of December 15, 2019, GEDmatch has not as yet prevented the attacks we describe. Since our and *Ney et al. (2020)*'s initial communications with GEDmatch in July, GEDmatch has placed a reCAPTCHA on its upload and 1-to-1 tool forms. Though reCAPTCHA may deter bulk bot attacks to harvest large numbers of kit genotypes, it is still possible for a human to carry out small-scale attacks. Further, even as reCAPTCHA has improved at blocking non-human users in recent years, new attacks have been developed to bypass reCAPTCHA (*Baecher et al., 2011*; *Brown et al., 2017*; *Zhou et al., 2018*; *Akrout et al., 2019*). As we outline below, there are simple steps that could be taken to make IBS attacks much less of a risk.

In our estimation, the other active services listed in *Table 1* (MyHeritage, FamilyTreeDNA, and LivingDNA) are likely substantially less vulnerable than GEDmatch to the attacks we describe here. LivingDNA does not provide a chromosome browser, precluding IBS tiling attacks. MyHeritage and FamilyTreeDNA use thresholds for revealing matching segment locations that make IBS tiling much less efficient. (However, FamilyTreeDNA's practice of showing matches as short as 1 cM given that two people share at least one long match is still somewhat permissive, see *Figure 2—figure supplement 2*). Representatives of MyHeritage, FamilyTreeDNA, and LivingDNA have confirmed to us that their IBD-calling algorithms rely on phased data, which should preclude IBS baiting. (We have not tested this ourselves). DTC genetic genealogy is a growing field, and any new entities that begin offering upload services may also face threats of the kind we describe.

Genetic genealogy databases that allow uploads have been in the public eye recently because of their role in long-range familial search strategies recently adopted by law enforcement. In long-range familial search, investigators seek to identify the source of a crime-scene sample by identifying relatives of the sample in a genetic genealogy database that allows uploads. Searching in SNP-based genealogy databases allows the detection of much more distant relationships than does familial searching in traditional forensic microsatellite datasets (*Rohlfs et al., 2012*), vastly increasing the number of people detectable by familial search (*Erlich et al., 2018*; *Edge and Coop, 2019*). At this writing, both GEDmatch and FamilyTreeDNA have been searched in this way. Long-range familial search raises a range of privacy concerns (*Syndercombe Court, 2018*; *Ram et al., 2018*; *Kennett, 2019*; *Scudder et al., 2019*). One response from advocates of long-range search has been to note that 'raw genetic data are not disclosed to law enforcement... Search results display only the length and chromosomal location of shared DNA blocks' (*Greytak et al., 2018*). However, the methods we describe here illustrate that there are several ways to reveal users' raw genetic data on the basis of the locations of shared DNA blocks. Because companies that work with law enforcement on long-range familial searching—including Parabon Nanolabs and Bode Technology (*Kennett, 2019*)—now routinely upload tens of datasets to genetic genealogy databases, they may be accidentally accumulating information that would allow them to reconstruct many people's genotypes.

Data breaches via IBS tiling, IBS probing, and IBS baiting are preventable. We have identified a set of strategies that genetic genealogy services could adopt to protect their genotype data from IBS-based attacks. We give a detailed list of these strategies in Appendix A (also summarized in *Table 2*). Broadly, the suggestions consist of restrictions on they types of datasets that can be uploaded, restrictions on the kinds of information shared with users, and restrictions on classes of methods used for identifying putative IBD segments. For example, to prevent IBS tiling, the simplest measures are either to forgo the use of a chromosome browser feature or only to show users the positions of long IBS segments, such as segments of at least 8 cM. To prevent IBS baiting, the most robust countermeasure is to phase data before identifying IBS segments, allowing only relatively few phase switches in any putative segment. Phasing the data and only reporting long segments both decrease the uncertainty of IBD calls and so may improve user experience as well. Finally, we also support the strategy of requiring encrypted signatures on uploaded files, proposed by *Erlich et al.*

**Table 2.** Potential countermeasures against the methods of attack outlined here, with their likely effectiveness against IBS tiling, IBS probing, and IBS baiting.

| Strategy | Prevents IBS tiling | Prevents IBS probing | Prevents IBS baiting |
|---|---|---|---|
| Require cryptographic signature from genotyping service | Yes | Yes | Yes |
| Restrict reporting of IBS to long segments (e.g. >8 cM) | Partially | Partially | Weakly |
| Report number and lengths of IBS segments but not locations | Yes | No | Partially |
| Block homozygous uploads | Partially | No | No |
| Report small number of matching individuals per kit | Partially | Partially | Partially |
| Disallow matching between arbitrary kits | Partially | Partially | Partially |
| Block uploads of publicly available genomes | Partially | No | No |
| Block uploads with evidence of IBS-inert segments | No | Yes | No |
| Block uploads with long runs of heterozygosity | No | No | Partially |
| Use phase-aware methods for IBS detection | No | No | Yes |

*(2018)*, which would allow DTC databases to block any files that do not originate from trusted sources. Some of our suggestions limit the potential uses of genetic genealogy data, and users will vary in the degree to which they value these potential uses and in the degree to which they want to protect their genetic information.

All these suggestions assume that genealogy services will maintain raw genetic data for people in their database. Another possibility would be for individual people instead to upload an encrypted version of their genetic data, with relative matching performed on the encrypted datasets, as has been suggested previously (*He et al., 2014*).

Our IBS tiling and IBS probing results focus on users of European ancestries, in part because most users of DTC genetic genealogy services appear to have substantial European ancestries. (DTC genetics companies generally do not release this kind of information on their users, but their research papers suggest that they have access to especially large samples with European ancestries—for example, a 23andMe paper on demography in the United States included almost 150,000 self-described European Americans and less than 10,000 each of self-described African Americans and Latino Americans [*Bryc et al., 2015*]. For a qualitatively similar sample composition in a study from Ancestry, see *Han et al. (2017)*. One question is how these results would generalize to other populations. Because IBD sharing is generally greater within populations than between populations (e.g. *Ralph and Coop, 2013*), potential users are more vulnerable if there are more publicly available genomes from people with similar ancestries. If IBD-detection algorithms are not well calibrated to differences in heterozygosity across populations, then spurious IBD calls will be more common in populations with lower heterozygosity, leading to greater risk of IBD tiling. Finally, we show in *Figure 2—figure supplement 4* that in our sample, Finnish samples are more vulnerable to IBS tiling than other populations, which is likely due to Finns tracing substantial ancestry to a founder population that experienced a bottleneck 100 generations ago (*Kere, 2001*). Members of other groups with similar demographic histories are likely to be at elevated risk of IBS tiling and IBS probing as well.

We have focused on genetic genealogy databases that allow uploads because at this writing, it is straightforward to download publicly available genetic datasets and to produce fake genetic datasets for upload. In principle, however, another way to perform attacks like the ones we describe would be to use biological samples. For example, a group of people willing to share their genetic data with each other could collaborate to perform IBS tiling by sending actual biological samples for genotyping. Even IBS probing and IBS baiting could be performed with biological samples by adversaries who could synthesize the samples. Though synthesizing such samples is technically challenging now, it may become easier in the future. Such methods could present opportunities to attack databases that do not allow uploads, such as the large databases maintained by Ancestry (>14 million) and 23andMe (>9 million) (*Regalado, 2019*). They would also thwart the countermeasure of requiring uploaded datasets to include an cryptographic signature indicating their source.

The IBS-based privacy attacks we describe here add to a growing set of threats to genetic privacy (*Homer et al., 2008*; *Nyholt et al., 2009*; *Im et al., 2012*; *Gymrek et al., 2013*; *Humbert et al.,*

*2015*; *Shringarpure and Bustamante, 2015*; *Edge et al., 2017*; *Ayday and Humbert, 2017*; *Kim et al., 2018*; *Erlich et al., 2018*). A person's genotype includes sensitive health information that might be used for discrimination, and people whose genetic information is compromised may be vulnerable to scams involving falsified relatives (*Ney et al., 2020*). Although there are many emerging threats to privacy, some of the more unsettling of which have nothing to do with genetics, genetic data do have special features that might require special considerations. In particular, genetic privacy concerns not only the person whose genotypes are directly revealed but also their relatives whose genotypes may be revealed indirectly (*Humbert et al., 2013*), a point highlighted by the use of genetic genealogy for long-range forensic searches (*Erlich et al., 2018*; *Edge and Coop, 2019*).

Although many forms of genetic discrimination are prohibited legally, rules vary between countries and states. For example, in the United States, the Genetic Information Nondiscrimination Act (GINA) protects against genetic discrimination in the provision of health insurance but does not explicitly disallow genetic discrimination in the provision of life insurance, disability insurance, or long-term care insurance (*Bélisle-Pipon et al., 2019*). In addition to measures for protecting genetic privacy in the short term, there is a need for more complete frameworks governing the circumstances under which genetic data can be used (*Clayton et al., 2019*).

## Materials and methods

### Data assembly

We performed IBS tiling with publicly available genoytpes from 872 people of European ancestries. Of these 872 genotypes, 503 came from the EUR subset of the 20130502 release of phase 3 of the 1000 Genomes project (*Abecasis et al., 2012*), downloaded from ftp://ftp.1000genomes.ebi.ac.uk/vol1/ftp/release/20130502/. This release set has been pruned to remove close biological relatives. The EUR subset includes the following population codes and numbers of people: CEU (Utah residents with Northern and Western European Ancestry, 99 people), FIN (Finnish in Finland, 99 people), GBR (British in England and Scotland, 91 people), IBS (Iberian Population in Spain, 107 people), TSI (Toscani in Italia, 107 people).

The remaining 369 were selected from samples typed on the Human Origins SNP array (*Patterson et al., 2012*), including 142 genotypes from the Human Genome Diversity Project (*Cann et al., 2002*). Specifically, we downloaded the Human Origins data from https://reich.hms.harvard.edu/downloadable-genotypes-present-day-and-ancient-dna-data-compiled-published-papers, using the 1240K+HO dataset, version 37.2. The 372 selected people were all contemporary samples chosen according to population labels. We also excluded people from the Human Origins dataset if they appeared in the 1000 Genomes dataset. The populations used for selecting data, along with the number of participants included after excluding 1000 Genomes samples, were as follows: 'Adygei' (16), 'Albanian' (6), 'Basque' (29), 'Belarusian' (10), 'Bulgarian' (10), 'Croatian' (10), 'Czech' (10), 'English' (0), 'Estonian' (10), 'Finnish' (0), 'French' (61), 'Greek' (20), 'Hungarian' (20), 'Icelandic' (12), 'Italian_North' (20), 'Italian_South' (4), 'Lithuanian' (10), 'Maltese' (8), 'Mordovian' (10), 'Norwegian' (11), 'Orcadian' (13), 'Romanian' (10), 'Russian' (22), 'Sardinian' (27), 'Scottish' (0), 'Sicilian' (11), 'Spanish' (0), 'Spanish_North' (0), and 'Ukrainian' (9). The populations with 0 people included are those for which all the samples in the Human Origins dataset are included in the 1000 Genomes phase 3 panel. Samples with group labels marked 'ignore' were excluded, including samples marked as close relatives.

We down-sampled the sequence data from the 1000 Genomes project to include only sites typed by the Human Origins chip. Of the 597,573 SNPs included in the Human Origins dataset, 558,257 sites appeared at the same position in the 1000 Genomes dataset, 557,999 of which appear as biallelic SNPs. For 546,530 of these, both the SNP identifier and position match in 1000 Genomes, and for 544,139 of them, the alleles agreed as well. We merged the dataset at the set of 544,139 SNPs at which SNP identifiers, positions, and alleles matched between the Human Origins and 1000 Genomes datasets.

We used vcftools (*Danecek et al., 2011*), bcftools (*Li, 2011*), PLINK (*Purcell et al., 2007*), and EIGENSOFT (*Price et al., 2006*) to create the merged file. The script used to create it is maintained at github.com/mdedge/IBS_privacy/, and the merged data file is available at https://doi.org/10.

25338/B8X619. A permanent version of the scripts used in the publication version of this paper is available with doi 10.5281/zenodo.3620958.

## Phasing, IBS calling, and IBS tiling

We phased the combined dataset using Beagle 5.0 (*Browning and Browning, 2007*) using the default settings and genetic maps for each chromosome. We used linear interpolation to obtain the genetic map position of each SNP on the build GRCh37 LDhat genetic map (*Frazer et al., 2007*) downloaded from the Beagle website (http://bochet.gcc.biostat.washington.edu/beagle/genetic_maps/). We used Refined IBD software (*Browning and Browning, 2013*) to identify IBS segments, retaining segments of at least .1 centiMorgans (cM) with LOD scores >1. We also used Germline (*Gusev et al., 2009*) to identify IBS segments under alternative parameters, shown in the supplement. The resulting IBS segments were analyzed using the GenomicRanges package (*Lawrence et al., 2013*) in R (*R Development Core Team, 2013*). Scripts used for phasing, IBS calling, and IBS tiling are available at github.com/mdedge/IBS_privacy/.

## IBS probing

To generate IBS-inert genotypes for IBS probing in *Figure 3*, we computed allele frequencies within the set of 872 Europeans for chromosome 19. Allele frequencies less than 10% were changed to 10%, and then alleles were sampled at one minus their frequency. This strategy generates genetic data that look quite unlike real data, with the advantage (for the purposes of IBS probing) of being unlikely to return IBS matches anywhere. An adversary attempting IBS probing in a real database would need to tailor the approach to the quality control and IBS calling methods used by the database.

After inert genotypes were produced, we stitched them with real phased genotypes from windows around GRCh position 45411941 on chromosome 19, the site of SNP rs429358. SNP rs429358 is in the APOE locus; if a haplotype has a C at rs429358 and a C at nearby SNP rs7412, then that haplotype is said to harbor the APO-$\epsilon 4$ allele, which confers risk for Alzheimer's disease (*Corder et al., 1993*). rs429358 is not genotyped on the Human Origins chip, but it is included on recent chips used by both Ancestry and 23andMe. To simulate probing with a 1 cM threshold for matching, we pulled real data from a region of 1.9 cM around the site, and to simulate probing with a 3 cM threshold, we pulled real data from a region of 5.9 cM around the site. Distances in cM were computed by linear interpolation from a genetic map in GRCh37 coordinates. Scripts used to generate *Figure 3* are available at github.com/mdedge/IBS_privacy/.

## GEDmatch demonstration

On Novermber 21st, 2019, we first uploaded artificial genetic datasets to GEDmatch's research mode in order to demonstrate the possibility of IBS baiting. GEDMatch has not published details of its IBS detection procedures. However, the options available to users in the 1-to-1 match tool and the description of how those options can be used to ignore single-site matches led us to hypothesize that GEDmatch uses phase-unaware IBS detection and that the 1-to-1 match tool might be vulnerable to IBS baiting.

### Description of GEDmatch 1-to-1 tool

GEDmatch's 1-to-1 match tool allows the user to compare the IBS matches of any two genetic datasets (or, in GEDmatch parlance, 'kits'), as long as the kit numbers are known to the user. Thus, to identify the genotypes of many users an adversary would need access to the kit numbers of many users. The 1-to-many tool in default GEDmatch reports 3000 of the closest genetic relatives of any kit whose number is known to the user, and reports the kit numbers of those match kits (along with names and email addresses). Thus an adversary can iteratively search for all the kit numbers matching a known kit, and so obtain many kit numbers to use in 1-to-1 searches. We alerted GEDmatch to this issue with the 1-to-many tool, as nearly the entire GEDmatch database of kit numbers and genetic relationships could be scraped.

The 1-to-1 match tool allows the user to specify parameters that govern IBS calling. In particular, the user can specify the minimum cM length of the blocks (down to 0.1 cM) and the minimum number of SNPs in a block (down to 25 SNPs). GEDmatch also allows the user to specify the 'mis-match bunch limit,' which appears to be the minimum number of IBS-compatible SNPs after an opposite-

homozygous site that are required in order for a second opposite-homozygous site not to break the IBS segment.

## Ethics

In order to comply with GEDmatch's terms and conditions, we used artificial datasets designed not to match any genuine genetic data uploaded to GEDmatch. The kits were uploaded in 'Research' mode, where they are not visible to other users via 1-to-many search. We did not interact with any other users' data; we ran GEDmatch's 1-to-1 comparison tools only comparing among our artificial kits. We exercised care not to interact with any other tools and to avoid accidental discoveries. Prior to uploading the artificial datasets, we also consulted with the UC Davis Institutional Review Board (IRB) to ensure that these uploads do not constitute human subjects research. Upon receiving confirmation from the IRB that our uploads do not constitute human subjects research and before uploading the datasets, we alerted GEDmatch that we would be making the uploads, and we also shared the kit numbers with them after we had completed our analyses.

## Construction of artificial datasets

We constructed artificial 'target' and 'bait' kits using the SNPs included in the 23 and Me v4 chip. (The 'target' kits are the targets of inference, and the 'bait' kits are designed to reveal their genotypes). We identified the alleles at these SNP positions in the 1000Genomes dataset, along with their frequencies in the EUR subset of 1000Genomes. We assigned as missing ('- -') any SNP that we could not match by position in 1000Genomes. We chose four target SNPs at random on chromosome 22. These SNPs were chosen at random from the set of strand-unambiguous polymorphisms, that is not A/T and G/C SNPs. These strand-unambiguous sites include the majority of SNPs on the chip, for example 89% of the SNPs on the 23andMe chip on chromosome 22.

### Target genomes

We uploaded three artificial target genome kits (T1-T3). These vary in their genotypes at the target SNPs. T1 and T3 are homozygous for different alleles; T2 is heterozygous. At the rest of the loci, we constructed genotypes by randomly sampling alleles according to their frequencies at each SNP. Thus, there is no LD among loci.

### Bait genomes

We uploaded two artificial bait genome kits. These two kits have opposite-homozygote genotypes at each of the target SNPs. The two bait uploads were then set to have identical genotypes in the rest of their autosomes, with their genotypes specified as below.

To create a region around the target that would bait a phase-unaware method into calling IBD, we took SNPs in the 0.6cM on either side of the target SNP, selected at random 22 on each side, and set them to be heterozygous in both bait genomes. The rest of the SNPs within this bait region were set to be missing. We used only 22 heterozygous SNPs on each side and filled in the rest with missing data (rather than making all sites heterozygous) because large numbers of heterozygous sites generated an error on upload, 'HTZ string too long' and would not be processed further. Blocking uploads with long runs of heterozygous sites is a countermeasure put in place by GEDmatch after we and (*Ney et al., 2020*) initially alerted GEDmatch to the risks of upload-based privacy attacks. However, we found that the countermeasure was not triggered by runs of heterozygous sites with missing sites interspersed, and these runs of heterozygosity interspersed with missingness also effectively baited GEDmatch into calling IBD segments. Additionally, we confirmed with Peter Ney (personal communication) that his previously uploaded kits including long runs of heterozygosity remain active even though re-uploads of those same kits are blocked as of December 3rd, 2019, suggesting that the block applies only to newly uploaded kits and not to existing data on GEDmatch.

The alleles in target kits at all other autosomal SNPs in the genome were drawn at random with frequency $1 - p$, where $p$ is the frequency in the 1000Genomes EUR subsample. This scheme was chosen to ensure that the bait genomes were unlikely to have spurious IBS matches anywhere with any target genome, so that the only potential IBS was in the target regions.

## Detailed results of baiting

We compared each target to both bait genomes using the 1-to-1 GEDmatch tool. We set the minimum block to a length of > 0.7cM and 25 SNPs, with a mismatch cutoff of 25 SNPs. This ensured that we could detect IBS in the key regions, but that a single opposite-homozygous mismatch would be sufficient to prevent the identification of a putative IBS segment in the key region.

The baiting attempt was successful; we observed IBS only where we expected it between bait and target kits (*Figure 5*). We observed no putative IBD segments on any chromosome except 22, as expected on the basis of our procedure for filling in artificial genotypes in both sets of kits. The details of the matches on chromosome 22 are reported in *Table 3*. We observed 4 putative IBD segments overlapping our target bait regions in the comparisons with matching homozygote genotypes at the bait site, that is in the T1-B1 and T3-B2 comparisons, as well in both heterozygote-homozygote comparisons, that is T2-B1 and T2-B2. We observed no putative IBD segments in the pairs with opposite-homozygous mismatches, T1-B1 and T3-B2. Thus the genotypes of the targets are readily discernable from from the putative IBD segments output by GEDmatch. The full results returned by

**Table 3.** Summary of the SNPs targeted by baiting and the IBS returned by GEDmatch.
For each region, we give the position of the key SNP (target bp). Because by design our bait kits are genetically identical outside of the target SNPs, the IBS regions returned by GEDmatch's 1-to-1 tool are identical across bait kits generating a match. For each pairwise comparison, we report the IBS information returned: Left-Right bp of the IBS region, the cM length, the number (#) of SNPs in the IBS region with a non-missing target. We also report the number (#) of SNPs spanned by the region IBS when matched to the missing target Bmiss.

| Matching pairs | | Target 1 | Target 2 | Target 3 | Target 4 |
|---|---|---|---|---|---|
| | target bp | 27613130 | 34024097 | 37673781 | 42008068 |
| T1-(B1 Bmiss) | | | | | |
| | IBS L bp | 27427698 | 33771672 | 37519864 | 40054428 |
| | IBS R bp | 27680780 | 34328741 | 37827711 | 43112674 |
| | IBS cM | 1.3 | 0.8 | 1.1 | 1.2 |
| | # SNPs | 47 | 45 | 42 | 40 |
| | # SNPs Bmiss | 46 | 44 | 41 | 39 |
| T2-(B1 B2 Bmiss) | | | | | |
| | IBS L bp | 27433179 | 33771672 | 37508507 | 40357667 |
| | IBS R bp | 27680780 | 34328741 | 37827711 | 43112674 |
| | IBS cM | 1.3 | 0.8 | 1.2 | 0.9 |
| | # SNPs | 45 | 45 | 45 | 32 |
| | # SNPs Bmiss | 44 | 44 | 44 | 31 |
| T3-(B3 Bmiss) | | | | | |
| | IBS L bp | 27433179 | 33771672 | 37519864 | 40357667 |
| | IBS R bp | 27680780 | 34328741 | 37827711 | 43112674 |
| | IBS cM | 1.3 | 0.8 | 1.1 | 0.9 |
| | # SNPs | 45 | 45 | 45 | 32 |
| | # SNPs Bmiss | 44 | 44 | 41 | 31 |
| Tmiss-(All Baits) | | | | | |
| | IBS L bp | 27433179 | 33771672 | 37519864 | 40357667 |
| | IBS R bp | 27680780 | 34328741 | 37827711 | 43112674 |
| | IBS cM | 1.3 | 0.8 | 1.1 | 0.9 |
| | # SNPs | 44 | 44 | 44 | 31 |
| | # SNPs Bmiss | 44 | 44 | 44 | 31 |

The online version of this article includes the following source data for  Table 3:
Source data 1. GEDmatch demonstration summary.

GEDmatch are available as images here (https://github.com/mdedge/IBS_privacy/tree/master/IBS_baiting_demo; the kit numbers are redacted to prevent reuse).

Some of the IBS blocks have fewer SNPs than we expect. We believe this to be due to the removal of SNPs during the tokenization stage, during which rare SNPs and SNPs with stand-ambiguous alleles seem to be removed (Ney et al., 2020). We did not investigate this further, but multiple uploads could be used to determine the approximate criteria for SNPs to be included, and hence determine where an adversary should set cutoffs.

Our two bait kits could both generate IBS matches to the target because the target genotype is missing rather than heterozygous. To determine whether a genotype was missing, we implemented a trick borrowed from Ney et al. (2020), and uploaded a third bait kit (Bmiss) with the target SNP set to missing (i.e. '- -') and then looked at the number of SNPs an IBS match across the target site spans. In each case, the non-missing baits (B1 and B2) generated an IBS block match with with T1-T3 that was one SNP longer than the IBS block generated by the Bmiss bait (Table 3). Comparing these baits to a new target with a missing genotype at each target site (Tmiss), we see that in each pairwise comparison the IBS blocks are the same number of SNPs long regardless of whether the target SNP bait genotype was missing (Table 3). Therefore, we can distinguish the target being heterozygote or missing by the use of a third bait kit and inspection of the number of SNPs included in an IBS match.

The possibility of IBS-baiting-like procedures also interacts with the vulnerabilities arising from the presentation of SNP-level visualizations explored by Ney et al. (2020). Even if short IBS blocks were not reported to the user explicitly, it is clear from the zoomed-in view that we can see the target mismatches in question (see Figure 5). One measure that GEDmatch appears to have taken against a Ney et al. (2020)-style attack is to jitter the positions of SNPs in their visualization slightly. However, an attacker could counter such jittering by embedding key sites in runs of heterozygosity, making it easier to identify them in visualizations after jittering. Thus, the images displayed by GEDmatch still pose additional security risks.

## Acknowledgements

We thank Matt Bishop, Elizabeth Joh, Peter Ney, and Mike Sweeney for useful conversations, and we thank Shai Carmi, Yaniv Erlich, Debbie Kennett, Leah Larkin, Magnus Nordborg, Rori Rohlfs, Noah Rosenberg, Ann Turner, Amy Williams, and an anonymous reviewer for helpful comments on the manuscript. Swapan Mallick and David Reich answered questions about the Human Origins dataset, Brian Browning answered questions about Refined IBD, and Alexander Gusev answered questions about Germline software. We acknowledge support from the National Institutes of Health (R01-GM108779 and F32-GM130050).

## Additional information

### Competing interests
Graham Coop: Reviewing editor, *eLife*. The other author declares that no competing interests exist.

### Funding

| Funder | Grant reference number | Author |
| --- | --- | --- |
| National Institutes of Health | GM108779 | Graham Coop |
| National Institutes of Health | GM130050 | Michael D Edge |

The funders had no role in study design, data collection and interpretation, or the decision to submit the work for publication.

### Author contributions
Michael D Edge, Graham Coop, Conceptualization, Resources, Data curation, Formal analysis, Funding acquisition, Validation, Investigation, Visualization, Methodology

## Author ORCIDs

Michael D Edge (ID) https://orcid.org/0000-0001-8773-2906
Graham Coop (ID) https://orcid.org/0000-0001-8431-0302

## Decision letter and Author response

Decision letter https://doi.org/10.7554/eLife.51810.sa1
Author response https://doi.org/10.7554/eLife.51810.sa2

## Additional files

### Supplementary files

- Transparent reporting form

### Data availability

The dataset used here was assembled from publicly available datasets. The combined dataset has been deposited in Dryad at https://doi.org/10.25338/B8X619, and scripts for assembling and analyzing the data are available at https://github.com/mdedge/IBS_privacy. Previously published data used includes the 1000 Genomes Project Phase 3 data (ftp://ftp.1000genomes.491ebi.ac.uk/vol1/ftp/release/20130502/) and downloadable genotypes of present-day and ancient DNA data (compiled from published papers) from the Reich Lab Harvard Medical School (https://reich.hms.harvard.edu/downloadable-genotypes-present-day-and-ancient-dna-data-compiled-published-papers).

The following dataset was generated:

| Author(s) | Year | Dataset title | Dataset URL | Database and Identifier |
|---|---|---|---|---|
| Michael D Edge, Graham Coop | 2020 | Data from: Attacks on genetic privacy via uploads to genealogical databases | https://doi.org/10.25338/B8X619 | Dryad Digital Repository, 10.25338/B8X619 |

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

## Appendix 1

# Detailed rationale for proposed countermeasures

Here, we detail the rationale and possible advantages and disadvantages of the countermeasures listed in *Table 2*.

1. Require uploaded files to include cryptographic signatures identifying their source.
   This recommendation was initially made by *Erlich et al. (2018)*. Under this suggestion, DTC genetics services would cryptographically sign the genetic data files they provide to users. Upload services might then check for a signature from an approved DTC service on each uploaded dataset, blocking datasets from upload otherwise. An alternative procedure that would accomplish the same goal would be for the DTC entities to exchange data directly at the user's request (*Ney et al., 2018*). Such a procedure would allow upload services to know the source of the files they analyze and to disallow uploaded datasets produced by non-approved entities and user-modified datasets. All the methods we describe require the upload of multiple genetic datasets. As such, requiring cryptographic signatures would force the adversary to have multiple biological samples analyzed by a DTC service in order to implement any of our procedures, and IBS probing and IBS baiting would require synthetic samples, which are much harder to produce than fake datasets. Another benefit of this approach is that it would protect research participants against being reidentified using DTC genetic genealogy services (*Erlich et al., 2018*). A disadvantage of this strategy is that it requires the cooperation of several distinct DTC services.

2. Restrict reporting of IBS to long segments.
   Reporting short IBS segments increases the typical coverage of IBS tiling (*Figure 2*) and IBS probing (*Figure 3*), as well as the efficiency of IBS baiting. Very short blocks may be of little practical utility for genetic genealogy (*Huff et al., 2011*). Reporting only segments longer than 8 cM would substantially limit IBS tiling attacks. A partially effective variant of this strategy is to report short segments only for pairs of people who share at least one long segment (*Figure 2—figure supplement 2*). One disadvantage is that short segments, though less reliably inferred than longer segments, may still be of interest to genealogists.

3. Do not report locations of IBS segments.
   Another tactic for preventing IBS tiling is not to report chromosomal locations at all. If chromosomal locations are not reported, IBS tiling as we have described it becomes impossible.

4. Block uploads of genomes with excessive homozygosity. IBS tiling is especially informative if genotypes that are homozygous for phased haplotypes are uploaded, so blocking genomes with excessive homozygosity presents a barrier to IBS tiling attacks. However, runs of homozygosity occur naturally (*Pemberton et al., 2012*), and allowing for naturally occurring patterns of homozygosity would leave a loophole for an adversary who could upload many genotypes, using including homozygous regions and using only those for tiling.

5. Report only a small number of putative relatives per uploaded kit.
   Reporting only the closest relatives (say the 50 - 100 closest relatives) of an uploaded kit would decrease the efficiency of all the methods we describe here. Only a small number of people could have their privacy compromised by each upload. This countermeasure comes with costs to genealogists, who may want to explore as many matches as possible in order to build family trees.

6. Disallow arbitrary matching between kits.
   Some services allow searches for IBS between any pair of individuals in the database. Allowing such searches makes all potential IBS attacks easier. This countermeasure might hamper the investigations of genealogists exploring complex hypotheses about relatedness.

7. Block uploads of publicly available genomes.
   There are now thousands of genomes available for public download, and these publicly available genomes can be used for IBS tiling. Genetic genealogy databases could include publicly available genomes (potentially without allowing them to be returned as IBS matches for typical users) and flag accounts that upload them. This strategy would go some distance toward blocking IBS tiling, but it could be thwarted in several ways, for example by uploading genetic datasets produced by splicing together haplotypes from publicly available genomes.

8. Block uploads with evidence of IBS-inert segments.
   IBS-inert segments—that is false genetic segments designed to be unlikely to be IBS with

anyone in the database—are key to IBS probing. Some methods for constructing IBS-inert segments are easy to identify, but others may not be. If a database is large enough, genomes with IBS-inert segments could be identified by looking for genomes that have much less apparent IBS with other database members than might be expected.

9. Block uploads with long runs of heterozygosity.
Long runs of heterozygosity do not arise naturally but are key to the IBS baiting approaches we describe here. Blocking genomes with long runs of heterozygosity—or alternatively, blocking genomes that have much more apparent IBS with a range of other database members than expected—would hamper IBS baiting. However, this countermeasure might be hard to apply to a small-scale IBS baiting attack, where only one or a few short runs of heterozygosity might be necessary. In our sample, the longest run of heterozygosity (in terms of number of SNPs) consisted of 38 SNPs and spanned .06 cM. This suggests that filtering out long runs of heterozygosity might be a promising strategy, though identifying a specific procedure would require more careful consideration of variation in non-European populations and of the composition of commercial SNP chips (including SNP density and allele frequencies).

10. Use phase-aware methods for IBS detection.
Although calling IBS by looking for long segments without incompatible homozygous genotypes scales well to large datasets, such methods are easy to trick, allowing IBS baiting approaches. In addition to allowing IBS estimation methods that are harder to trick, faked samples may stand out as unusual during the process of phasing, raising more opportunities for quality-control checks.

