## [Decision Letter]

**Acceptance summary:**

The manuscript shows that genetic databases, like other databases, are vulnerable to being (ab)used in a manner not foreseen by their owners. The presented scenarios are very realistic and I sincerely hope that this article will spur both genetic service firms and politics into action.

**Decision letter after peer review:**

Thank you for submitting your article "Attacks on genetic privacy via uploads to genealogical databases" for consideration by *eLife*. Your article has been reviewed by three peer reviewers, and the evaluation has been overseen by Magnus Nordborg as the Reviewing Editor and Mark McCarthy as the Senior Editor. The following individuals involved in review of your submission have agreed to reveal their identity: Amy L Williams (Reviewer #1); Shai Carmi (Reviewer #2). The reviewers have discussed the reviews with one another and the Reviewing Editor has drafted this decision to help you prepare a revised submission.

Summary:

Direct-to-consumer genetics services are increasingly popular for genetic genealogy, with millions of customers as of 2019. Several DTC genealogy services allow users to upload their own genetic data in order to search for genetic relatives. This paper demonstrates that such services can also be exploited to reveal much about the genome of individuals in the database, without their consent, and with potentially adversarial consequences.

Essential revisions:

A major concern was whether you intend this paper as a hypothetical discussion, or as a real-world demonstration. The latter would require real-world examples. Either way, it would be necessary to try to specify the conditions under which this work. For example, your IBD baiting method depends sensitively on the algorithms used by the provider. Thus, at a minimum, we would like to see a general discussion of the limitations of these exploits, and of the parameters that effect their efficacy.

Needless to say, you should also discuss the recent work of Ney et al., which you are aware of.

Reviewer #1:

Edge and Coop outline several means of inferring genotypes of individuals in genetic databases that allow users – specifically an attacker – to upload genetic data. There are three such attacks: (1) IBS tiling, wherein an attacker utilizes inferred IBS regions to N (known) samples to reconstruct the genotypes of others in the database in a genome-wide fashion. (2) IBS probing, with the aim of inferring genotypes at a specific locus (for example, the APOE locus). And (3) IBS baiting, with a more systematic form of probing that, given multiple uploads, may be able to collect enough genotypes to allow imputation of a person's common variants genome-wide.

The paper is of broad interest given the robust interest in direct-to-consumer genetic testing and the advent of long-range familial searches by law enforcement. The authors have responsibly handled their discoveries, notifying the various companies that host databases vulnerable to these attacks.

Before it can be published a few technical topics that should be addressed are:

1) To perform IBS baiting effectively, the (unphased) IBS detector would need to break segments at opposite homozygous genotypes. However, in practice most such detectors would likely be tolerant to errors, and therefore may not break the segments in the fashion the text assumes. This seems like it may pose a serious issue to such an attack, and this merits more careful consideration.

2) The Supplementary Figure 4 legend says, "All other arguments were kept at their default values. Calling IBS without respect to genotype phase returns many IBS segments, but less can be learned about each segment via tiling than if haplotype phase is respected." Note that services that I'm aware of do not provide phase to the users, nor allow users to upload phased data. (Please say if this is not the case.) Given this, it's hard to make use of the potential benefits of phased data. In general, this issue merits a more prominent mention – the fact that, if phase were known, the amount of information learned by an attacker is greater, and that (unless services give this information), it's hard to envision how an attacker could put this into practice.

3) "(We consider some alternative IBS reporting procedures in the supplement.)" Is this about the supplementary figures? If so, perhaps cite them. I wasn't sure if there was text that I missed?

4) Optionally, the Discussion may wish to expand on the proposals that inhibit genealogical research, which end users may prefer not to have implemented. Points 2, 4, and 5 (since inferring other relatives' relationships can be useful) fall in this category.

Reviewer #2:

Conflict of interest:

The authors have shared the manuscript with me a few weeks ago. I have discussed my thoughts with the authors and they have revised their manuscript according to some of my comments, acknowledging me at the end of the paper. The review below is a slightly edited and expanded version of the remaining comments.

General assessment:

Edge and Coop describe a method for breaching privacy of genomes deposited in genetic genealogy matching services. The key idea is that a genetic match between a (known) uploaded genome and a target genome reveals some DNA sequence of the target. This can be exploited to recover large proportions of the target's genome. This is a novel and innovative approach. The manuscript is very interesting and thought provoking and the results are important. The analysis is overall sound (but see the points below). I am certain that the results will have major implications on genetic genealogy, genetic privacy, and beyond.

1) A result I find intriguing is the high proportion of the genome that can be covered by IBS tiling, which is comparable only to what was previously seen in founder populations. I suggest the following. (All the requested data should be already available to the authors or require very quick experiments.)

a) Show a breakdown of the coverage by ethnicity (at least for the main ethnicities), to make the results a bit more comparable to previous studies.

b) Further emphasize the message that segments need not be IBD (identity-by-descent) to allow privacy breaching – rather, IBS (identity-by-state) is good enough. This explains to some extent why the proportions of genome covered are so high, even though these are (mostly) not founder populations.

c) I think the results of Figure 2—figure supplement 3 are important, and should perhaps be reported as the main results. The reason is that when running Germline in haploid mode with no errors allowed, we are guaranteed no mismatches between the target and the (known) uploaded genome. In other words, we have an exact match to at least one haplotype of the target. (The authors can even improve performance easily by using a diploid mode but allowing no errors. Germline would still require a perfect match, but would allow phasing errors if they happen between blocks.) If the authors choose not to change the order of figures, I would recommend to at least report the mismatch rate between the haplotypes that were found to be matching by RefinedIBD in the main analysis (Figure 2).

d) Regarding Supplementary Figure 4, I think this figure might be somewhat misleading. The problem with the approach taken to generate that figure (if I'm not mistaken) is that Germline will not try to match any sites where either the target or the uploaded genome are heterozygous. Thus, the coverage is likely inflated – there could be entire "covered" segments that provide very little information on the target. At the very least, the uploaded genome should be made artificially homozygous, so that we are guaranteed to have information on the homozygous genotypes of the target.

2) It will be important to evaluate the IBS tiling method against a very simple "null", in which each allele is predicted to be the major (more frequent) allele. In other words, the outcome would be not the proportion of the genome covered, but the proportion of alleles of the target correctly inferred, and this outcome should be compared between IBS tiling and just using the major allele. While this experiment may take a little time (but I believe no more than a few days), I believe it is essential, because otherwise it is difficult to evaluate the success of the proposed method.

3) I am not confident whether these very elegant results form a practical and immediate risk of privacy, or whether the paper is more of a proof of concept. The biggest problem is with IBS baiting. The success of this approach relies on an IBD detection algorithm that is very simple minded. It is not clear to me whether any of the companies is actually using such an algorithm. But more generally, the authors did not demonstrate an actual recovery of genomic material from a genetic genealogy service using any of their methods. Of course, they would not want, and should not, violate the terms and conditions of any company. But I think that if using research genomes (such as 1000 Genomes) or their own genomes, and limiting the experiment in duration and scale, this would be legitimate. Or the authors could even explicitly ask the companies' managements for permission.

This is not to say that the article is not worthwhile without such experiments. On the contrary, the paper describes a very novel approach, and it would be extremely important and urgent that the proposed techniques become known to all stakeholders in personal genomics, both from the industry and from the academia, as well as the actual participants. Also, additional experiments may take too much time or be outside the scope of the present paper.

But as happens frequently with this kind of papers, once they are published, the media and the general public cannot get to the bottom of such subtle nuances (even if authors do their very best). I expect the paper will be very widely covered, and with some likelihood, it could develop into a total media circus and trigger panic. I think that would be an unfortunate consequence, unless there is a real, tangible risk of privacy breaching. If the risk is more theoretical in nature, it will be important to say so explicitly (and possibly drop the part about the letters to the representatives of the companies, which is only going to amplify the drama).

Reviewer #3:

Edge and Coop describe a battery of methods that seek to recover parts of a personal genome through segment matching queries in a direct-to-consumer database that facilitates uploads. Specifically, the authors describe methods for tiling the hacked genome with matched segments, probing it for the genotype at a particular locus, or baiting it to match contrived genomes, designed to recover the genotype at a particular site.

The paper is technically sound. Methodologically, it puts together ideas that had been floated, and actually evaluates them rigorously. In the context of the genetic privacy field, this constitutes and advancement.

1) This reviewer believes that genetic privacy as a whole is overblown. The impacts of violating it are not substantial, and accepting such work in broadly read venues panders to irrational fears thus does science a disservice. While I don't fault the authors for pushing their work to a visible journal, making this more of a comment to the editor, I would nevertheless welcome the authors' rebuttal. Specifically, I would challenge the statements in the last paragraph of the Discussion regarding trait-predictability of traits. These are upper-bounded by the prediction accuracy implied by SNP heritability (accuracy which is markedly lower than the SNP heritability itself). More practically, the likely improvement in prediction does not mean convergence of prediction even to that bound. Worse, given the non-genetic data trace of individuals today, with more precious predictive value, genetic privacy is a distraction. An example ad absurdum, every street camera recovers my height better than my genome would.

2) The paper is somewhat thin in results (basically, Figures 2 and 3). In particular, Section 2.3 is falsely appearing under Results, whereas it only describes a method, without even applying it. This defeats the entire purpose of the manuscript, of actually demonstrating the attacks and quantifying their effectiveness. One quantitative question relevant to (defending against) the baiting attack has to do with feasibility of assembling all-het segments from naturally-occurring human haplotypes of chip SNPs. There are back-of-an-envelope reasons to assume those would not be long enough for the described attack, but actual data would be reassuring and consistent with the nature of contributions of this manuscript.

3) Relatedly, I am specifically concerned regarding the baiting security loophole being practical, as the authors' description of IBS baiting relies on a straw man IBS detector that they construct to have that weakness. As the authors point out, many actual detectors would not willy-nilly extend each segment till conflicting homozygous on both ends, or require some information content to seed a match between segments. Baiting may still be possible, but likely more complicated and potentially impractical.

4) The results reported are all w.r.t. the general European population. It is important to report the (different) results for other continental ancestries, and, on the other hand, in bottleneck populations.

---

## [Author Response]

Essential revisions:A major concern was whether you intend this paper as a hypothetical discussion, or as a real-world demonstration. The latter would require real-world examples. Either way, it would be necessary to try to specify the conditions under which this work. For example, your IBD baiting method depends sensitively on the algorithms used by the provider. Thus, at a minimum, we would like to see a general discussion of the limitations of these exploits, and of the parameters that effect their efficacy.

The reviewers’ comments in this area focus on IBS baiting. We have clarified that IBS baiting is possible in GEDmatch (as of Nov and Dec 2019) in two ways: (1) Section VII of Ney et al., which became public shortly after we submitted, executes an attack essentially equivalent to IBS baiting in section VII, and (2) We have performed IBS baiting in a small set of fake genetic datasets that we uploaded to GEDmatch in research mode. (These uploads do not violate GEDmatch’s terms and were determined not to be human subjects research by our university IRB. We also confirmed our plan with GEDmatch before initiating uploads).

We have also expanded the discussion of the circumstances that would make the attacks harder or easier, clarifying the implications with respect to the various existing genealogy services.

Needless to say, you should also discuss the recent work of Ney et al., which you are aware of.

Yes, we have added discussion of Ney et al., which dovetails well with our results, at various relevant points. We mention this in the manuscript, but we want to state clearly here that the work of Ney et al. was entirely independent of ours; we were unaware of their work until after our manuscript was submitted to *eLife* and bioRxiv. Ney et al. described an exploit on the images provided by GEDmatch’s 1-to-1 search feature. We pointed out that these images were a major security weakness to GEDmatch in our July 24^th^ email to them (all our emails to GEDmatch are now posted on the paper’s GitHub repository), but we did not pursue them further. The other exploit discussed by Ney et al. operates on the same principles as our IBS baiting procedure (described in their section VII).

Reviewer #1:[…] The paper is of broad interest given the robust interest in direct-to-consumer genetic testing and the advent of long-range familial searches by law enforcement. The authors have responsibly handled their discoveries, notifying the various companies that host databases vulnerable to these attacks.Before it can be published a few technical topics that should be addressed are:1) To perform IBS baiting effectively, the (unphased) IBS detector would need to break segments at opposite homozygous genotypes. However, in practice most such detectors would likely be tolerant to errors, and therefore may not break the segments in the fashion the text assumes. This seems like it may pose a serious issue to such an attack, and this merits more careful consideration.

We agree that this is an important issue for planning a method of attack. On the basis of our exploration of GEDmatch’s features, we have always believed that it is susceptible to an IBD baiting attack. The flexibility of their 1-to-1 interface allows users to effectively force IBS blocks to be broken by single mismatches, as the user gets to choose the tolerance to errors, by specifying the number of matching SNPs before a second mismatch is allowed. Indeed the video that GEDmatch points users to demonstrates how to vary the tool’s parameters to allow single mismatches to be ignored (https://www.youtube.com/watch?v=7J2TGtcOYMs&feature=youtu.be). We had initially not gone ahead with the demonstration on GEDmatch as we had wanted to avoid getting into detailed explorations of specific databases, and we initially also did not want to single out the only non-commercial database for a demonstration of an attack.

The Ney et al. paper (https://dnasec.cs.washington.edu/genetic-genealogy/ney_ndss.pdf) independently arrived at an attack that is essentially equivalent to IBS baiting as we describe it (with the modification of adding an additional file that reveals whether the site is missing in the target) and showed that it works in GEDmatch. The attack is described in section VII of their paper.

We have now demonstrated that IBS bating works in GEDmatch. We first found this in late November, and we confirmed that the methods still work as of December 15th, after the acquisition of GEDmatch by Verogen. We have described these in depth in sections 2.3.3 and 4.4. We note that in response to our and Ney et al.’s initial emails GEDmatch had put in place a number of counter-measures to try and block IBS-baiting style attacks, notably blocks on long runs of heterozygosity. However, it only took a few trial uploads to work out the rough parameters of these checks, and to design baiting kits that could circumvent their checks. GEDmatch has also put in place recaptcha on their 1-to-1 tool page to block bots, this may block simple bot attacks to access large numbers of genotypes.

We have currently chosen not to publicly release the code to simulate our GEDmatch baits, to make it slightly less trivial to perform the hack. The code is very simple. We are happy to reconsider this decision if the editor or reviewers feel that the code is needed.

We have also added a bit more general discussion on this issue in the IBS baiting section. The key point is that the attacker needs to design a set of uploads where changes in called IBS (or lack of change in called IBS) with a target will reveal the target’s genotype at a known site. If the adversary can identify the algorithm, it may often be possible to design a strategy that reveals many genotypes, albeit perhaps not all of them, and perhaps requiring more uploaded datasets. To illustrate this, we have added a short discussion of what one might do if the database allows one incompatible homozygote within any called IBD segment.

2) The Supplementary Figure 4 legend says, "All other arguments were kept at their default values. Calling IBS without respect to genotype phase returns many IBS segments, but less can be learned about each segment via tiling than if haplotype phase is respected." Note that services that I'm aware of do not provide phase to the users, nor allow users to upload phased data. (Please say if this is not the case.) Given this, it's hard to make use of the potential benefits of phased data. In general, this issue merits a more prominent mention – the fact that, if phase were known, the amount of information learned by an attacker is greater, and that (unless services give this information), it's hard to envision how an attacker could put this into practice.

(We have deleted the former Supplementary Figure 4 in response to reviewer 2’s) comments, but here is a response to the rest of the comment): We agree that a service that provides phasing information and allows phased uploads would be easier to exploit, and that most of the services currently do not, which we have clarified. (GEDmatch actually does seem to allow phased matches to be shown---there is a color on their one-to-one match pictures intended to show a phased match---but we do not believe they are implemented in much generality. Since they’re there, though, we’ve avoided saying that none of the services used phased inputs.) We also have emphasized that one way around this concern is to upload homozygous genotypes, and we have emphasized that Figure 2—figure supplement 3 is a projection of the success one might obtain by doing so (since phase errors break IBS segments in Germline’s haploid mode). Filters can be set up to prevent uploads with unrealistic amounts of homozygosity, but an adversary could upload genomes with homozygous runs in key places that are not homozygous everywhere. Because long runs of homozygosity appear naturally, it may be difficult to filter out datasets with enough homozygous material to be useful for IBS tiling.

3) "(We consider some alternative IBS reporting procedures in the supplement.)" Is this about the supplementary figures? If so, perhaps cite them. I wasn't sure if there was text that I missed?

Yes, we meant the supplementary figures and should have been clearer about that. We have deleted this line from the Discussion and clarified the referencing of the supplementary figures in the Results section.

4) Optionally, the Discussion may wish to expand on the proposals that inhibit genealogical research, which end users may prefer not to have implemented. Points 2, 4, and 5 (since inferring other relatives' relationships can be useful) fall in this category.

Yes, we have added some short notes about the potential costs to legitimate genealogical pursuits of these countermeasures. (In response to a comment from reviewer 3, we have moved the detailed discussion of these points to the Appendix.)

Reviewer #2:[…] Edge and Coop describe a method for breaching privacy of genomes deposited in genetic genealogy matching services. The key idea is that a genetic match between a (known) uploaded genome and a target genome reveals some DNA sequence of the target. This can be exploited to recover large proportions of the target's genome. This is a novel and innovative approach. The manuscript is very interesting and thought provoking and the results are important. The analysis is overall sound (but see the points below). I am certain that the results will have major implications on genetic genealogy, genetic privacy, and beyond.1) A result I find intriguing is the high proportion of the genome that can be covered by IBS tiling, which is comparable only to what was previously seen in founder populations. I suggest the following. (All the requested data should be already available to the authors or require very quick experiments.)a) Show a breakdown of the coverage by ethnicity (at least for the main ethnicities), to make the results a bit more comparable to previous studies.

We have added a supplementary figure (Figure 2—figure supplement 4) showing coverage for four EUR subgroups in 1000 Genomes, namely FIN (Finnish), IBS (Iberian in Spain), GBR (British), and TSI (Tuscany). We chose these four subgroups of the combined dataset because they are all large enough subsets to get decent estimates, they are reasonably clearly localized geographically, and they are of approximately equal size. As might be expected, the Finnish sample shows higher coverage by IBS tiling than other groups. We have added some comments about variation among groups in the success of IBD tiling to the Discussion.

b) Further emphasize the message that segments need not be IBD (identity-by-descent) to allow privacy breaching – rather, IBS (identity-by-state) is good enough. This explains to some extent why the proportions of genome covered are so high, even though these are (mostly) not founder populations.

Thank you. We have emphasized this point by adding these sentences to section 2.1: “True IBD segments reveal more than mere IBS segments about shared genotypes because untyped variants (including rare variants) within an IBD segment are likely to be shared. At the same time, mere IBS is sufficient to infer sharing for SNPs that are genotyped within the segment.”

c) I think the results of Figure 2—figure supplement 3 are important, and should perhaps be reported as the main results. The reason is that when running Germline in haploid mode with no errors allowed, we are guaranteed no mismatches between the target and the (known) uploaded genome. In other words, we have an exact match to at least one haplotype of the target. (The authors can even improve performance easily by using a diploid mode but allowing no errors. Germline would still require a perfect match, but would allow phasing errors if they happen between blocks.) If the authors choose not to change the order of figures, I would recommend to at least report the mismatch rate between the haplotypes that were found to be matching by RefinedIBD in the main analysis (Figure 2).

We agree with this concern but have chosen to retain the figure order. The main reason is that during revision, we realized we had misread GEDmatch’s minimum cM threshold, which is in fact 0.1cM and not 1cM. Because of this, we wanted to add results for 0.1cM, but Germline’s options do not allow the user to search for such short IBS segments (perhaps reasonably for most applications), and so we stuck with refinedIBD for the main text. Still, we have added comments to the main text to emphasize the point raised by the reviewer and point readers to Figure 2—figure supplement 3.

d) Regarding Supplementary Figure 4, I think this figure might be somewhat misleading. The problem with the approach taken to generate that figure (if I'm not mistaken) is that Germline will not try to match any sites where either the target or the uploaded genome are heterozygous. Thus, the coverage is likely inflated – there could be entire "covered" segments that provide very little information on the target. At the very least, the uploaded genome should be made artificially homozygous, so that we are guaranteed to have information on the homozygous genotypes of the target.

We agree that the kind of coverage tracked by the former Supplementary Figure 4 is not very informative about the target for the reasons stated---a tile only reveals that the target is unlikely to have a homozygote opposite to the comparison within the region. We have removed it for now, though reviewer 1 appeared to find the figure informative, and we are open to reintroducing it with extra emphasis in the caption on the limited information gained from tiles in this case.

2) It will be important to evaluate the IBS tiling method against a very simple "null", in which each allele is predicted to be the major (more frequent) allele. In other words, the outcome would be not the proportion of the genome covered, but the proportion of alleles of the target correctly inferred, and this outcome should be compared between IBS tiling and just using the major allele. While this experiment may take a little time (but I believe no more than a few days), I believe it is essential, because otherwise it is difficult to evaluate the success of the proposed method.

We have added a supplementary figure that addresses this concern (Figure 2—figure supplement 5). We are not 100% sure of the alternative hypothesis being proposed and have not included a hypothesis test. (Does the alternative hypothesis allow prediction of the major allele outside IBS tiles? It seems to us that would be the most natural comparison.) But the figure supplement shows the median proportion of total alleles covered and the median proportion of minor alleles covered (minor alleles are ~19% of the total). The figure supplement suggests that there is a slight bias for IBS tiles to be in regions of lower SNP density and in regions with lower heterozygosity. However, the biases are relatively small, so, for example, with a 1cM threshold and all 872 samples included, IBS tiles cover a median of 52% of the minor alleles (as opposed to 57% of the total length in base pairs). Though we do not give these numbers in text, one can calculate easily from the figure legend that a guess of major alleles everywhere gives an average ~81% of alleles guessed accurately (the average major allele frequency), whereas tiling plus a guess of major alleles outside of the tiles gives ~91% accuracy for the median person (all the major alleles plus about half the minor alleles), an increase that is sure to be significant with the many loci considered.

3) I am not confident whether these very elegant results form a practical and immediate risk of privacy, or whether the paper is more of a proof of concept. The biggest problem is with IBS baiting. The success of this approach relies on an IBD detection algorithm that is very simple minded. It is not clear to me whether any of the companies is actually using such an algorithm. But more generally, the authors did not demonstrate an actual recovery of genomic material from a genetic genealogy service using any of their methods. Of course, they would not want, and should not, violate the terms and conditions of any company. But I think that if using research genomes (such as 1000 Genomes) or their own genomes, and limiting the experiment in duration and scale, this would be legitimate. Or the authors could even explicitly ask the companies' managements for permission.This is not to say that the article is not worthwhile without such experiments. On the contrary, the paper describes a very novel approach, and it would be extremely important and urgent that the proposed techniques become known to all stakeholders in personal genomics, both from the industry and from the academia, as well as the actual participants. Also, additional experiments may take too much time or be outside the scope of the present paper.But as happens frequently with this kind of papers, once they are published, the media and the general public cannot get to the bottom of such subtle nuances (even if authors do their very best). I expect the paper will be very widely covered, and with some likelihood, it could develop into a total media circus and trigger panic. I think that would be an unfortunate consequence, unless there is a real, tangible risk of privacy breaching. If the risk is more theoretical in nature, it will be important to say so explicitly (and possibly drop the part about the letters to the representatives of the companies, which is only going to amplify the drama).

We thank reviewer 2 for this thoughtful comment. We have made several changes to address it. First, we note that section VII of the Ney paper already shows that IBS baiting has recently been possible in GEDmatch.

As discussed in the response to reviewer 1 we have now implemented a demonstration of IBS baiting against GEDmatch, and we present the results in sections 2.3.3 and 4.4. We have also added more discussion on the specific risks at the services listed in Table 1. Our general view is that the risks are substantial at GEDmatch and much lower (but still likely nonzero) at the other services.

Reviewer #3:[…] 1) This reviewer believes that genetic privacy as a whole is overblown. The impacts of violating it are not substantial, and accepting such work in broadly read venues panders to irrational fears thus does science a disservice. While I don't fault the authors for pushing their work to a visible journal, making this more of a comment to the editor, I would nevertheless welcome the authors' rebuttal. Specifically, I would challenge the statements in the last paragraph of the Discussion regarding trait-predictability of traits. These are upper-bounded by the prediction accuracy implied by SNP heritability (accuracy which is markedly lower than the SNP heritability itself). More practically, the likely improvement in prediction does not mean convergence of prediction even to that bound. Worse, given the non-genetic data trace of individuals today, with more precious predictive value, genetic privacy is a distraction. An example ad absurdum, every street camera recovers my height better than my genome would.

We agree with many of the points raised here and have expanded the paragraph mentioned to emphasize the points of agreement. In particular, we agree that for many complex traits, prediction may be bounded at fairly low accuracy, even as sample sizes and models improve, and we have dropped the sentence noted by the reviewer. We also agree that there are many other threats to privacy, such as street cameras and many kinds of traces of our behavior online. These threats are much more revealing about many aspects our lives than genetics (and doubtless this will remain true into the future). At the same time, one does not need to adopt a strong version of genetic exceptionalism to be concerned about a new form of data being leaked. As geneticists we view it as important to highlight issues with genetic data.

These databases are growing rapidly and have garnered a lot of public attention, both through advertising by companies and news stories following the Golden State Killer. The fact that the genetic data of over a million people, notably through GEDmatch, may have been open to an adversarial attack does concern us and seems worthy of public attention. One reason for choosing *eLife* as a venue for the article is that we wanted the article to be fully open access so that the large communities of genetic genealogists could discuss the final version of paper (and freely reuse figures etc.).

Genetic data privacy and sharing are evolving rapidly, and if no clear thinking on policy emerges, it will be too late to reverse decisions that have been made by default. The potential risks from genetic data breaches extend beyond the targeted person, at least to the target’s relatives. Further, others have also noted that there are potential national security issues arising from genetic data breaches (e.g. identification of covert operatives), and that people whose genetic information is compromised might be vulnerable to cyberattacks (e.g. an attacker might generate a genetic profile for a false relative to gain a person’s trust).

It may well be that in the long run, “genetic privacy” is not the right framing for these issues, and that we may need to move to a framework of ownership of genetic data, allowed uses of genetic data, and harms done by misuse. Still, we believe that making the potential risks clear as quickly as possible is a useful contribution to the public discussion.

2) The paper is somewhat thin in results (basically, Figures 2 and 3). In particular, Section 2.3 is falsely appearing under Results, whereas it only describes a method, without even applying it. This defeats the entire purpose of the manuscript, of actually demonstrating the attacks and quantifying their effectiveness. One quantitative question relevant to (defending against) the baiting attack has to do with feasibility of assembling all-het segments from naturally-occurring human haplotypes of chip SNPs. There are back-of-an-envelope reasons to assume those would not be long enough for the described attack, but actual data would be reassuring and consistent with the nature of contributions of this manuscript.

In response to this and the other reviewer comments, we have added an application of IBS baiting in GEDmatch, similar to the analysis of Ney et al. section VII (which was performed independently and without our knowledge).

We have also added the following text after identifying the longest run of heterozygosity (in terms of # of SNPs) in our dataset: “In our sample, the longest run of heterozygosity (in terms of number of SNPs) consisted of 38 SNPs and spanned. 06 cM. This suggests that filtering out long runs of heterozygosity might be a promising strategy, though identifying a specific procedure would require more careful consideration of variation in non-European populations and of the composition of commercial SNP chips (including SNP density and allele frequencies).”

3) Relatedly, I am specifically concerned regarding the baiting security loophole being practical, as the authors' description of IBS baiting relies on a straw man IBS detector that they construct to have that weakness. As the authors point out, many actual detectors would not willy-nilly extend each segment till conflicting homozygous on both ends, or require some information content to seed a match between segments. Baiting may still be possible, but likely more complicated and potentially impractical.

We agree that the IBS detector we consider is simplistic, but as mentioned above, IBS baiting does work in GEDmatch, indicating that the IBS detection scheme we describe is not far off of the actual method used in a major database. In response to this comment and one of reviewer 1’s comments, we have added some text on how one might respond to slightly more sophisticated IBS callers in section 2.3.1. It seems to us that there will be some potential to violate privacy using a baiting-like procedure in any database that uses phase-unaware IBS detection and does not take steps to filter out fake profiles. It may become impractical at some point, but it is better to make it effectively impossible by using phase-aware IBS detection.

4) The results reported are all w.r.t. the general European population. It is important to report the (different) results for other continental ancestries, and, on the other hand, in bottleneck populations.

We have added some discussion of the differences among some of the European subgroups (see also the new supplementary figure, Figure 2—figure supplement 4), which allows some extrapolation to patterns that will affect other continental groups. We have chosen to focus on European populations in part because major genetic genealogy databases seem to consist largely of people of European ancestries. We have added the following paragraph to the Discussion:

“Our IBS tiling and IBS probing results focus on users of European ancestries, in part because most users of DTC genetic genealogy services appear to have substantial European ancestries. […] Finally, we show in Figure 2—figure supplement 4 that in our sample, Finnish samples are more vulnerable to IBS tiling than other populations, which is likely due to Finns tracing substantial ancestry to a founder population that experienced a bottleneck 100 generations ago (Kere, 2001). Members of other groups with similar demographic histories are likely to be at elevated risk of IBS tiling and IBS probing as well.”